# Neural Collapse beyond the Unconstrained Features Model: Landscape, Dynamics, and Generalization in the Mean-Field Regime

Diyuan Wu [1]   Marco Mondelli [1]

## Abstract

Neural Collapse is a phenomenon where the last-layer representations of a well-trained neural network converge to a highly structured geometry. In this paper, we focus on its first (and most basic) property, known as NC1: the within-class variability vanishes. While prior theoretical studies establish the occurrence of NC1 via the data-agnostic unconstrained features model, our work adopts a data-specific perspective, analyzing NC1 in a three-layer neural network, with the first two layers operating in the mean-field regime and followed by a linear layer. In particular, we establish a fundamental connection between NC1 and the loss landscape: we prove that points with small empirical loss and gradient norm (thus, close to being stationary) approximately satisfy NC1, and the closeness to NC1 is controlled by the residual loss and gradient norm. We then show that *(i)* gradient flow on the mean squared error converges to NC1 solutions with small empirical loss, and *(ii)* for well-separated data distributions, both NC1 and vanishing test loss are achieved simultaneously. This aligns with the empirical observation that NC1 emerges during training while models attain near-zero test error. Overall, our results demonstrate that NC1 arises from gradient training due to the properties of the loss landscape, and they show the co-occurrence of NC1 and small test error for certain data distributions.

## 1. Introduction

Neural Collapse (NC), first identified by Papyan et al. (2020), describes a phenomenon observed during the final stages of training where: *(i)* the penultimate-layer features converge to their respective class means (**NC1**), *(ii)* these class means form an equiangular tight frame (ETF) or an orthogonal frame (**NC2**), and *(iii)* the columns of the final layer's classifier matrix similarly form an ETF or orthogonal frame, implementing a nearest class-mean decision rule on the penultimate-layer features (**NC3**). A popular line of theoretical research has investigated the occurrence of NC via the unconstrained features model (UFM), see (Fang et al., 2021a; Han et al., 2022; Mixon et al., 2022) and the discussion in Section 2. In this framework, the penultimate-layer features are treated as free optimization variables, leading to a benign loss landscape for the resulting optimization problem. The primary justification for adopting the UFM is that the complex feature-learning layers encountered in practice are approximated by a universal learner. While the UFM provides an intriguing theoretical perspective on NC, it has notable limitations. In particular, it neglects the dependence on the data distribution, rendering it unsuitable for theoretically analyzing the relationship between NC during training and the test error (Hui et al., 2022). Furthermore, the training dynamics under the UFM framework is not equivalent to the actual training dynamics of neural networks, which makes it challenging to investigate the occurrence of NC from a dynamical perspective.

To address the limitations of UFM, we consider training a three-layer network via gradient flow on the standard mean squared error (MSE) loss. Specifically, we employ a two-layer neural network in the mean-field regime (Mei et al., 2018) as the feature-learning component, and then concatenate it with a linear layer as the final predictor. Our main results both *(i)* establish sufficient conditions on the loss landscape for the first – and most basic – property of neural collapse, i.e., NC1, to hold, and *(ii)* show that such conditions are in fact satisfied by training the architecture above. This differentiates our paper from recent studies aiming to theoretically explain the NC phenomenon beyond unconstrained features, as existing work either provides only sufficient conditions for NC to occur (Seleznova et al., 2024), focuses on the NTK regime (Jacot et al., 2024), relies on specific training algorithms (Beaglehole et al., 2024) or on a specific regularization (Hong & Ling, 2024a), see Section 2 for a discussion of related work. Specifically, our

---

[1]Institute of Science and Technology Austria (ISTA), Klosterneuburg, Austria. Correspondence to: Diyuan Wu <diyuan.wu@ist.ac.at>, Marco Mondelli <marco.mondelli@ist.ac.at>.

*Proceedings of the 42nd International Conference on Machine Learning*, Vancouver, Canada. PMLR 267, 2025. Copyright 2025 by the author(s).

contributions are summarized below:

- First, we connect the emergence of NC1, i.e., the fact that the within-class variability vanishes, with properties of the loss landscape: we show that all approximately stationary points with small empirical loss are roughly NC1 solutions, and the degree to which the within-class variability vanishes is controlled by gradient norm and loss. This implies the prevalence of NC1 during training, as practical training procedures typically converge to such points with small gradient and loss.

- Next, we prove that gradient flow on a three-layer network operating in the mean-field regime satisfies the two conditions above (small gradient norm and small loss) and, therefore, it converges to an NC1 solution. While achieving approximately stationary points is expected, the primary challenge lies in controlling the empirical loss due to the model's non-convex nature.

- Finally, we show that, for certain well-separated data distributions, it is possible to achieve NC1 during training as well as vanishing test error, which corroborates the empirical finding that NC1 and strong generalization occur simultaneously.

## 2. Related Work

**Neural collapse: UFM and beyond.** The introduction of the UFM in (Mixon et al., 2022; Fang et al., 2021a) has prompted a line of work studying the emergence of neural collapse for that model. Specifically, Zhou et al. (2022) focus on the two-layer UFM model, showing that all its stationary points satisfy neural collapse. Han et al. (2022) prove convergence of gradient flow on UFM to NC solutions. Tirer & Bruna (2022) demonstrate that the global minimizers also satisfy neural collapse when the UFM has multiple linear layers or it incorporates the ReLU activation. Súkeník et al. (2023) extend the results to a deep UFM model for binary classification. Súkeník et al. (2024) then show that, for the deep UFM and multi-class classification, all the global optima still satisfy NC1, but not NC2 and NC3, due to the low-rank bias of the model. We also refer to (Kothapalli, 2023) for a rather recent and detailed review.

Going beyond the UFM, Seleznova et al. (2024) study the connection between NC and the neural tangent kernel (NTK), showing NC under certain block structure assumptions on the NTK matrix. However, the occurrence of such a block structure during training is unclear. Beaglehole et al. (2024) establish NC both empirically and theoretically for Deep Recursive Feature Machine training – a method that constructs a neural network by iteratively mapping the data through the average gradient outer product and then applying an untrained random feature map. Pan & Cao (2023) consider classification with cross-entropy loss, providing a quantitative bound for NC. Kothapalli & Tirer (2024) focus

on two-layer neural networks in both the NNGP and the NTK limit, proving neural collapse for $1$-dimensional Gaussian data. Hong & Ling (2024a) study NC for shallow and deep neural networks, also characterizing the generalization error. However, they regularize the loss by the $L_2$-norm of the features rather than the weights, which is different from the weight decay used in practice. Jacot et al. (2024) establish the occurrence of NC for deep neural networks with multiple linear layers, given a balancedness assumption on all the linear layers; in addition, they also prove that balancedness is achieved via gradient descent training using NTK tools. Compared to (Jacot et al., 2024), our proof does not rely on any balancedness condition, and it only requires the gradient norm to be small, which is naturally achievable via gradient flow. In fact, the stationary points to which our results apply may not be balanced, see the discussions at the end of Section 4.1 and 4.2.

**Mean-field analysis for networks with more than two layers.** While the properties of the loss landscape and training dynamics of two-layer neural networks in the mean-field regime have been extensively studied (Mei et al., 2018; Chen et al., 2020; Javanmard et al., 2020; Shevchenko et al., 2022; Hu et al., 2021; Suzuki et al., 2024a; Takakura & Suzuki, 2024), networks with more than two layers still prove to be challenging to analyze. Prior works (Lu et al., 2020; Araújo et al., 2019; Shevchenko & Mondelli, 2020; Fang et al., 2021b; Pham & Nguyen, 2021; Nguyen & Pham, 2023) have investigated the mean-field regime for deep neural networks, where the widths of all layers tend to infinity. In contrast, we let only the width of the first layer tend to infinity, while the width of the second layer remains of constant order. A closely related paper is by Kim & Suzuki (2024), which studies the in-context loss landscape of a two-layer linear transformer with a formulation similar to ours. However, the global convergence results in (Kim & Suzuki, 2024) rely on assumptions such as absence of weight decay, taking a two time-scale limit, and using a birth-death process (rather than the widely-used gradient flow), which are not applicable to our setting.

## 3. Problem Setting

**Notation.** Given an integer $n$, we use the shorthand $[n] := \{1, \ldots, n\}$. Given a vector $v \in \mathbb{R}^d$, let $v[i]$ be its $i$-th entry and $\text{Diag}(v) \in \mathbb{R}^{d \times d}$ the diagonal matrix with $v$ on the diagonal. Let $\mathbf{1}_d \in \mathbb{R}^d$ be the all-one vector of dimension $d$. Given a matrix $A$, let $[A]_{i,j}$ be its $(i,j)$-th element. We denote by $\|\cdot\|_F, \|\cdot\|_{op}$ the Frobenius and operator norms of a matrix, and by $\langle A, B \rangle_F = \text{Tr}\{A^\top B\}$ the Frobenius inner product. Let $\otimes$ be the Kronecker product and $vec(\cdot)$ the vectorization of the matrix obtained by stacking columns. Given a vector valued function $f : \mathbb{R}^d \to \mathbb{R}^d$, we denote by $\nabla \cdot f : \mathbb{R}^d \to \mathbb{R}$ its divergence. Given a real valued function $g$, we denote by $\|g\|_\infty = \sup |g|$ its infinity

norm. Let $\mathscr{P}_2(\mathbb{R}^d)$ be the space of probability measures on $\mathbb{R}^d$ with finite second moment, and $\mathcal{W}_1(\cdot,\cdot), \mathcal{W}_2(\cdot,\cdot)$ the Wasserstein-1 and -2 metrics, respectively.

**Three-layer fully connected neural networks.** We start by defining the following infinite-width neural network as a feature-learning layer:

$$h_\rho(x) = \mathbb{E}_\rho[a\sigma(u^\top x)], \tag{1}$$

where $a \in \mathbb{R}^p, u, x \in \mathbb{R}^d$, and $\rho = Law(a, u)$ is the joint measure of $a, u$. The network is parameterized by a probability distribution $\rho \in \mathscr{P}_2(\mathbb{R}^{p+d})$, and it represents the mean-field limit of the finite-width two-layer network below (Mei et al., 2018):

$$h_N(x) = \frac{1}{N}\sum_{j=1}^N a_j\sigma(u_j^\top x), \ \ \theta_j = (a_j, u_j) \overset{i.i.d.}{\sim} \rho. \tag{2}$$

For technical convenience, throughout this paper we directly consider the infinite-width network (1). In fact, its difference with the finite-width counterpart (2) can be readily bounded using results from (Mei et al., 2018; 2019).

Next, we cascade a linear layer, obtaining a three-layer neural network as follows:

$$f(x; \rho, W) = \gamma W^\top h_\rho(x), \tag{3}$$

where $W \in \mathbb{R}^{p \times q}$ and $\gamma \in \mathbb{R}$ is a (constant) multiplicative factor. Throughout the paper, we will refer to $f$ in (3) as the predictor. Neural networks with two linear layers in the end are also studied by Jacot et al. (2024), and adding a multiplicative factor $\gamma$ is proposed by Chen et al. (2020) to guarantee the global convergence of the dynamics.

The motivation for considering this model is to explore how training data affects the emergence of neural collapse. While previous studies on UFM offer insights into neural collapse, their key limitation lies in the disregard for the influence of training data. The primary justification for using the UFM is that it functions as a universal learner, thus emulating the complex feature learning layers encountered in practice. The three-layer network in the mean-field regime defined in (3) not only performs feature learning by taking into account the training data, but the feature layer in (1) is also recognized as a universal learner (Ma et al., 2022).

*q***-class balanced classification.** We consider a $q$-class balanced classification problem, with each class having $m$ data points. We denote by $n = qm$ the total number of training samples and assume that $p \geq q$. The empirical loss function is given by

$$\mathcal{L}_n(\rho, W) = \frac{1}{2n}\|\gamma W^\top H_\rho - Y\|_F^2,$$

where

$$H_\rho = [h_\rho(x_1), \ldots, h_\rho(x_n)] \in \mathbb{R}^{p \times n},$$
$$Y = [\underbrace{e_1, \ldots e_1}_{m \text{ columns}}, \ldots, e_q, \ldots e_q] \in \mathbb{R}^{q \times n}.$$

We also denote $X = [x_1, \ldots, x_n] \in \mathbb{R}^{d \times n}$. We consider a regularized problem with $L^2$ and entropy regularization, denoting the regularized loss and the free energy as

$$\mathcal{L}_{\lambda,n}(\rho, W) = \mathcal{L}_n(\rho, W) + \frac{\lambda_W}{2}\|W\|_F^2 + \frac{\lambda_\rho}{2}\mathbb{E}_\rho[\|\theta\|_2^2], \tag{4}$$

$$\begin{aligned}\mathcal{E}_n(\rho, W) = \ &\mathcal{L}_n(\rho, W) + \frac{\lambda_W}{2}\|W\|_F^2 \\ &+ \frac{\lambda_\rho}{2}\mathbb{E}_\rho[\|\theta\|_2^2] + \beta^{-1}\mathbb{E}_\rho[\log\rho],\end{aligned} \tag{5}$$

where the entropic regularization is explicitly added to perform noisy gradient flow, see (20). While we focus on balanced classification for technical clarity and brevity, our results extend to unbalanced classification (as considered e.g. in (Thrampoulidis et al., 2022; Hong & Ling, 2024b)) and regression (as considered e.g. in (Andriopoulos et al., 2024)) with minimal modifications.

**Neural collapse metric.** We focus on the first property of neural collapse and, given a feature matrix $H \in \mathbb{R}^{p \times n}$, we consider the following metric of NC1 as the ratio between in-class variance and total variance:

$$NC1(H) = \frac{\text{Tr}\left\{(\widetilde{H} - M_c)^\top(\widetilde{H} - M_c)\right\}}{\text{Tr}\left\{\widetilde{H}^\top \widetilde{H}\right\}},$$

where $\widetilde{H} \in \mathbb{R}^{p \times n}$ is the matrix of centered features and $M_c \in \mathbb{R}^{p \times n}$ the matrix of in-class means, defined as

$$M_g = \frac{1}{n}H\mathbf{1}_n\mathbf{1}_n^\top, \ \ \widetilde{H} = H - M_g, \ \ M_c = \frac{1}{m}\widetilde{H}Y^\top Y.$$

In words, if $NC1(H)$ is small, the within-class variability is negligible compared to the overall variability across classes, capturing the closeness of features to respective class means.

## 4. Within-class Variability Collapse during Training

### 4.1. Sufficient Conditions for NC1

Throughout the paper, we make the following assumptions that are mild and standard in the related literature, see e.g. (Mei et al., 2018; Chen et al., 2020; Suzuki et al., 2024a).

**Assumption 1.** (A1) *Regularity of the initialization:* We initialize the training algorithm with $W_0$ such that $W_0^\top W_0 = I_q$ and $\rho_0 = \mathcal{N}(0, I_{p+d})$.

(A2) *Boundedness of the data:* for all $i$, $\|x_i\|_2 \leq 1$.

(A3) *Regularity of the activation function:* $\|\sigma(z)\|_\infty$, $\|\sigma'(z)\|_\infty$, $\|\sigma''(z)\|_\infty$, $\|\sigma'''(z)\|_\infty$, $\|(z\sigma'(z))'\|_\infty \leq C_1$ for some universal constant $C_1$.

We remark that the ReLU activation does not satisfy the regularity conditions above, but we still expect our results to hold by taking the limit of a sequence of approximations of ReLU.

We now define an $\epsilon_S$-stationary point of the free energy.

**Definition 4.1.** We say that $(\rho, W)$ is an $\epsilon_S$-stationary point of $\mathcal{E}_n(\rho, W)$ w.r.t. $\rho$ if the following holds:

$$\mathbb{E}_\rho\left[\left\|\nabla_\theta \frac{\delta}{\delta\rho}\mathcal{E}_n(\rho, W)\right\|_2^2\right] \le \epsilon_S^2.$$

Here, we recall that, given a functional $G : \mathscr{P}_2(\mathbb{R}^D) \to \mathbb{R}$, its first variation at $\rho$ is the function $\frac{\delta}{\delta\rho}G(\rho)(\cdot) : \mathbb{R}^D \to \mathbb{R}$ such that, for all $\rho' \in \mathscr{P}_2(\mathbb{R}^D)$,

$$\int \frac{\delta}{\delta\rho}G(\rho)(\theta)\,(\rho' - \rho)\mathrm{d}\theta = \lim_{\epsilon \to 0} \frac{G((1-\epsilon)\rho + \epsilon\rho') - G(\rho)}{\epsilon}.$$

The result below (proved in Appendix B.1) characterizes the feature $H_\rho$ at any $\epsilon_S$-stationary point.

**Theorem 4.2.** *Under Assumption 1, for any $\epsilon_S$-stationary point $(\rho, W)$, we have the following characterization of the learned feature:*

$$vec(H_\rho) = \left(\gamma \frac{K_\rho(X, X)}{n} \otimes W\right)$$
$$\cdot \left(\gamma^2 \frac{K_\rho(X, X)}{n} \otimes (W^\top W) + \lambda_\rho I_{nq}\right)^{-1} vec(Y)$$
$$+ \boldsymbol{E}_1(\epsilon_S, \lambda_\rho; \gamma, W),$$

(6)

*where*

$$\|\boldsymbol{E}_1(\epsilon_S, \lambda_\rho; \gamma, W)\|_2^2 \le 2\left(\lambda_\rho^{-4}\gamma^4 C_1^2 \sigma_{\max}(W)^4 + \lambda_\rho^{-2}\right)C_1^2 n\epsilon_S^2,$$

(7)

*and the kernel $K_\rho(X, X) \in \mathbb{R}^{n \times n}$ induced by $\rho$ is*

$$K_\rho(X, X) = \mathbb{E}_\rho[\sigma(X^\top u)\sigma(u^\top X)]. \quad (8)$$

*As a consequence, if $W$ is non-singular, we have*

$$H_\rho = \gamma^{-1}W(W^\top W)^{-1}Y + \boldsymbol{E}_2(\epsilon_S, \lambda_\rho; \gamma, \rho, W), \quad (9)$$

*where*

$$\|\boldsymbol{E}_2(\epsilon_S, \lambda_\rho; \gamma, \rho, W)\|_F^2 \le 2\left(\lambda_\rho^{-4}\gamma^4 C_1^2 \sigma_{\max}(W)^4 + \lambda_\rho^{-2}\right)$$
$$\cdot C_1^2 n\epsilon_S^2 + \frac{2n\gamma^{-2}\mathcal{L}_n(\rho, W) + 2\lambda_\rho^{-2}\sigma_{\max}(W)^2 C_1^2 n\epsilon_S^2}{\sigma_{\min}(W)^2}.$$

(10)

*Proof sketch.* As $(\rho, W)$ is $\epsilon_S$-stationary, the following expression for $a$ holds almost surely w.r.t. the measure $\rho$:

$$a + \frac{\gamma\lambda_\rho^{-1}}{n}W(\gamma W^\top H_\rho - Y)\sigma(X^\top u)$$
$$+ \lambda_\rho^{-1}\beta^{-1}\nabla_a \log \rho(\theta) = O(\epsilon_S), \quad (11)$$

where, with an abuse of notation, the term $O(\epsilon_S)$ indicates that the norm of the vector on the LHS is at most of order $\epsilon_S$. By plugging (11) into $H_\rho = \mathbb{E}_\rho[a\sigma(u^\top X)]$, we obtain

$$H_\rho = -\lambda_\rho^{-1}\gamma W(\gamma W^\top H_\rho - Y)\frac{K_\rho(X, X)}{n} + O(\epsilon_S). \quad (12)$$

Note that (12) is a linear equation in $H_\rho$. Thus, by solving it explicitly and tracking the error in $\epsilon_S$, we obtain (6). Finally, the crux of the argument for (9) is to use again stationarity to show that (up to an error of order $\epsilon_S$)

$$(K' \otimes W')\left(K' \otimes (W'^\top W') + \lambda_\rho I_{nq}\right)^{-1} vec(Y)$$
$$= vec(W'(W'^\top W)^{-1}Y) + O(\mathcal{L}_n(\rho, W)), \quad (13)$$

where $K' := K_\rho(X, X)/n$ and $W' := \gamma W$. □

Note that $\gamma^{-1}W(W^\top W)^{-1}Y$, i.e., the first term in the decomposition of $H_\rho$ in (9), satisfies NC1. Indeed, $Y$ is the one-hot vector of labels and, thus, for two data point $x_i, x_j$ in the same class $k$, we have $y_i = y_j$, which implies that $W(W^\top W)^{-1}y_i = W(W^\top W)^{-1}y_j$. The second term $\boldsymbol{E}_2$ in the decomposition (9) is small, as long as $\epsilon_S$ and $\mathcal{L}_n(\rho, W)$ are small. Hence, the key question is *whether we can achieve a nearly stationary point having a small loss $\mathcal{L}_n(\rho, W)$ via a certain training dynamics*, which is addressed in the next sections.

As the result in (9) requires $W$ not to be too ill-conditioned, we now prove that this is the case, as long as the regularization terms $\lambda_W, \lambda_\rho, \beta^{-1}$ and the regularized loss $\mathcal{L}_{\lambda,n}$ are sufficiently small.

**Lemma 4.3.** *Let $\lambda_\rho = \lambda_\rho^0\beta^{-1}$, $\lambda_W = \lambda_W^0\beta^{-1}$ where $\lambda_\rho^0, \lambda_W^0$ are universal constants. Fix any $\alpha \ge 0$, $0 < \epsilon_0 \le 1/2$, and assume that*

$$\beta \ge \max\left\{e^{\frac{4\alpha}{\epsilon_0}\log\frac{2\alpha}{\epsilon_0}}, e^{4\alpha\log(2\alpha)}, (2C_1^2 nB(\lambda_\rho^0)^{-1})^{\frac{2}{\epsilon_0}},\right.$$
$$\left.\left(\frac{4q}{n}\right)^{\frac{1}{\epsilon_0}}, 64(qB)^2\right\},$$

(14)

*for some constant $B$ that doesn't depend on $\beta$. Suppose further that $(\rho, W)$ is any point such that*

$$\mathcal{L}_{\lambda,n}(\rho, W) \le B\beta^{-1}(\log \beta)^\alpha. \quad (15)$$

*Then, we have that*

$$\sigma_{\min}(W) \ge \beta^{-\epsilon_0},$$
$$\sigma_{\max}(W)^2 \le \|W\|_F^2 \le 2B(\lambda_W^0)^{-1}(\log \beta)^\alpha.$$

The proof is by contradiction. Suppose that $W$ has a small singular value, then the projection of $H_\rho$ in the corresponding left singular space of $W$ needs to be large, since the

regularized loss is small and $Y$ is isotropic. However, large component of $H_\rho$ in a subspace will in turn lead to large regularized loss due to the second-moment regularization term. The complete argument is deferred to Appendix B.2.

Next, we compute the NC1 metric induced by Theorem 4.2.

**Corollary 4.4.** *Consider the setting of Theorem 4.2 and assume that*

$$\|\boldsymbol{E}_2\|_F^2 \leq \frac{1}{8\sigma_{\max}^2(W)} \frac{(q-1)n}{q}, \tag{16}$$

*where* $\boldsymbol{E}_2 := \boldsymbol{E}_2(\epsilon_S, \beta; \gamma, \rho, W)$ *is bounded as in (10). Then, for any* $\epsilon_S$-*stationary point* $(\rho, W)$ *with non-singular $W$, we have*

$$NC1(H_\rho) \leq \frac{16\|\boldsymbol{E}_2\|_F^2}{\frac{1}{2\sigma_{\max}^2(W)}\frac{(q-1)n}{q} - 4\|\boldsymbol{E}_2\|_F^2}. \tag{17}$$

The proof of Corollary 4.4 is a direct calculation, and it is provided in Appendix B.3. Note that, in the setting of Lemma 4.3, we have that

$$\frac{1}{8\sigma_{\max}^2(W)} \frac{(q-1)n}{q} = \Omega((\log \beta)^{-\alpha}),$$
$$\|\boldsymbol{E}_2\|_F^2 = \mathcal{O}(\beta^{-1+2\epsilon_0}(\log \beta)^\alpha + \beta^{4+2\epsilon_0}(\log \beta)^{4\alpha}\epsilon_S^2). \tag{18}$$

Now, let us pick a sufficiently small $\beta^{-1}$ (corresponding to small regularization) and then a sufficiently small $\epsilon_S$ (corresponding to reaching a stationary point). Then, (18) implies that (16) holds and the upper bound on the NC1 metric in (17) vanishes.

**Imbalancedness of stationary point.** The recent work by Jacot et al. (2024) shows that, for any network with at least two consecutive linear layers in the end, sufficiently small loss and approximate balancedness of the linear layers suffice to guarantee NC1. Our network defined in (3) has two final linear layers, but due to the entropic regularization, all stationary points of the free energy are not balanced, which means that the techniques in (Jacot et al., 2024) cannot be applied to our setup. To demonstrate this, we prove in Appendix B.4 the following result.

**Lemma 4.5.** *Let* $(\rho, W)$ *be a stationary point of the free energy, i.e.,*

$$\nabla_\theta \frac{\delta}{\delta\rho} \mathcal{E}_n(\rho, W) = 0, \quad \nabla_W \mathcal{E}_n(\rho, W) = 0,$$

*with* $\mathbb{E}_\rho[a] < \infty$. *Then, any stationary point satisfies*

$$\lambda_\rho \mathbb{E}_\rho[aa^\top] - \lambda_W WW^\top = \beta^{-1}I_p. \tag{19}$$

The result in (19) implies that the network cannot be balanced, i.e., $\mathbb{E}_\rho[aa^\top]$ cannot be proportional to $WW^\top$. In fact, assume that $\lambda_\rho$ and $\lambda_W$ are of same order as $\beta^{-1}$, i.e.,

$\lambda_\rho = \lambda_\rho^0 \beta^{-1}$ and $\lambda_W = \lambda_W^0 \beta^{-1}$ for universal constants $\lambda_\rho^0, \lambda_W^0$. Then, as $WW^\top$ is of rank $q < p$, (19) gives that, for any constant $c$, $\|\mathbb{E}_\rho[aa^\top] - cWW^\top\|_{op} \geq (\lambda_\rho^0)^{-1}$.

We complement the theoretical result in Lemma 4.5 with numerical simulations, discussed at the end of Section 4.2, showing that for there are settings such that gradient-based training over standard datasets (MNIST, CIFAR-100) the neural network achieves NC1 without converging to a balanced solution.

### 4.2. Achieving NC1 via Gradient-based Training

From Theorem 4.2 and Corollary 4.4, we know that NC1 is achieved at any $\epsilon_S$-stationary point w.r.t. $\rho$ having small empirical loss. We now consider training $\rho$ and $W$ with gradient flow, i.e.,

$$dW_t = -\nabla_W \mathcal{L}_{\lambda,n}(\rho_t, W_t)dt;$$
$$d\theta_t = -\nabla_\theta \frac{\delta}{\delta\rho} \mathcal{L}_{\lambda,n}(\rho_t, W_t)(\theta_t)dt + \sqrt{2\beta^{-1}}dB_t, \tag{20}$$

and we show that, having trained long enough, one ensures that both $\epsilon_S$ and the empirical loss are sufficiently small.

The convergence to an $\epsilon_S$-stationary point with arbitrary small $\epsilon_S$ is a direct consequence of the fact that, under gradient flow, the gradient norm vanishes.

**Lemma 4.6.** *Under Assumption 1, fix* $\beta, \gamma > 0$ *and consider an initialization* $(\rho_0, W_0)$ *with finite free energy. For* $t \geq 0$, *let*

$$\epsilon_S^t = \mathbb{E}_{\rho_t}\left[\left\|\nabla_\theta \frac{\delta}{\delta\rho} \mathcal{E}_n(\rho_t, W_t)(\theta_t)\right\|_2^2\right],$$

*which is equivalent to* $(\rho_t, W_t)$ *being an* $\epsilon_S^t$-*stationary point. Then, for any* $\epsilon_S > 0$, *there exists* $T(\epsilon_S) > 0$ *s.t. for all* $t > T(\epsilon_S)$ *except a finite Lebesgue measure set,*

$$\epsilon_S^t \leq \epsilon_S.$$

The proof of Lemma 4.6 is provided in Appendix C.1. This result directly implies that $\liminf_{t\to+\infty} \epsilon_S^t = 0$.

Next, Theorem 4.8 shows that, by picking large enough $\gamma$ and training long enough, we achieve $\mathcal{O}(\beta^{-1})$ empirical loss. This requires the following mild assumptions that imply the positive definiteness of the kernel $K_\rho$ in (8) at initialization, as showed in Lemma 4.7.

**Assumption 2.** Assume $\sigma^{(2k)} \neq 0$ for all $k > 0$ and that there exist $s \in [d]$ s.t. *(i)* $x_i[s] \neq x_j[s]$ for all $i \neq j \in [n]$, and *(ii)* $x_i[s] \neq 0$ for all $i \in [n]$.

In words, the activation function is required to be smooth and have non-zero even derivatives, which is satisfied by e.g. $\texttt{tanh}$ or the sigmoid (also fulfilling Assumption 1). As for the training data, we assume that it is non-degenerate and not parallel, which holds for most practical data sets.

The next technical lemma, which comes from (Nguyen & Mondelli, 2020, Lemma 3.4)[1], shows the required positive definiteness of the kernel.

**Lemma 4.7.** *Under Assumption 2, let* $K(X, X) = \mathbb{E}_{u \sim \gamma_d}[\sigma(X^\top u)\sigma(u^\top X)]$, *where* $\gamma_d = \mathcal{N}(0, I_d)$. *Then,*

$$\lambda_* := \lambda_{\min}(K(X, X)) > 0.$$

We are now ready to state our result showing the convergence of the empirical loss to a low loss manifold by running gradient flow for long enough time.

**Theorem 4.8.** *Let Assumptions 1, 2 hold, set* $\lambda_\rho = \lambda_W = \beta^{-1}$ *and*

$$\gamma > C_3, \quad t_0 = \beta C_5. \tag{21}$$

*Then, for any* $\beta$ *and any* $t \geq t_0$, *we have*

$$\mathcal{L}_{\lambda,n}(\rho_t, W_t) \leq \beta^{-1} C_4, \tag{22}$$

*where* $C_3, C_4, C_5$ *are constants that depends on* $n, d, p, C_1, \lambda_*$ *but not on* $\beta$.

The expression of $C_3, C_4, C_5$ is provided in Theorem C.1, whose statement and proof are in Appendix C.2. We note that the choice $\lambda_\rho = \lambda_W = \beta^{-1}$ is only for technical convenience, what matters here is that $\lambda_\rho, \lambda_W = \Theta(\beta^{-1})$.

*Proof sketch.* We start by defining a first-hitting time

$$t_* = \min\{\inf\{t : \|W_t^\top W_t - W_0^\top W_0\|_{op} > R_W\}, \\ \inf\{t : D_{KL}(\rho_t \| \rho_0) > R_\rho\}\}, \tag{23}$$

where $D_{KL}(\cdot \| \cdot)$ denotes the KL divergence. Intuitively, (23) means that, for $t < t_*$, the gradient flow stays in a ball around the initialization. The crux of the argument is to show that, with a suitable choice of $R_W, R_\rho$, the loss becomes small before the dynamics has exited the ball.

To do so, we first prove in Lemma C.2 that, for $t < t_*$,

$$\mathcal{L}_n(\rho_t, W_t) \leq \exp(-\gamma^2 A_1 t) + \gamma^{-2} \beta^{-2} A_2, \tag{24}$$

for some $A_1, A_2$ that do not depend on $\gamma, \beta$ (but only on $\lambda_*, p, d, n$). This implies that the empirical loss converges exponentially fast (in $t$) to an error of order $\beta^{-2}$, as long as the gradient flow is inside the ball. We then show that, by picking a proper $\gamma$, $t_*$ is large enough so that the first term in (24) is of order $\beta^{-2}$ for some $t_0 < t_*$.

Finally, by combining the upper bound on the empirical loss with the fact that $D_{KL}(\rho_{t_0}, \rho_0), \|W_{t_0}\|_F^2$ are bounded by a constant independent of $\beta$, we obtain that $\mathcal{E}_n(\rho_{t_0}, W_{t_0}) = \mathcal{O}(\beta^{-1})$. Thus, since the free energy decreases along the gradient flow, the upper bound in (22) follows from the relationship between empirical loss and free energy proved in Lemma A.3. □

---
[1] Note that the lemma is contained in the v1 of the paper, available on `arXiv`.

We remark that we also provide a convergence rate for the loss in (24). The problem is challenging due to its non-convex nature, and to our best knowledge, no explicit rate of convergence is known in the mean-field regime beyond the two-layer case (which has a convex free energy).

**Comparison with related work.** While the strategy described above is motivated by and similar to that used in (Chen et al., 2020, Theorem 4.4), its technical implementation differs, due to differences in the problem setting. In fact, Chen et al. (2020) consider two-layer neural networks whose free-energy landscape is strongly convex in $\rho$. In contrast, the presence of the last linear layer $W$ implies that the free-energy landscape is non-convex in $(\rho, W)$, which makes it less obvious that gradient flow converges to a low free-energy manifold. As a consequence, Chen et al. (2020) show that the gradient flow dynamics stays in a ball around initialization with a certain radius for infinitely long time. In contrast, in our case, the gradient flow dynamics stays in the ball only for finite time, but this finite time suffices to ensure a small enough free energy.

Finally, the combination of Theorem 4.8, Lemma 4.6 and Corollary 4.4 gives that NC1 provably holds under gradient flow training.

**Corollary 4.9.** *Consider the setting of Theorem 4.8 and, for any* $0 < \delta_0 < 1$, *let*

$$\beta > \max \left\{ (2C_1^2 n C_4)^6, \left(\frac{4q}{n}\right)^3, \right.$$
$$\left. 64(qC_4)^2, \left(640 C_3^{-2} C_4^3 \frac{1}{\delta_0}\right)^3 \right\},$$

*where* $\gamma, t_0$ *as chosen as in (21). Then, there exists* $T(\beta) > 0$ *s.t. for all* $t > \max\{T(\beta), t_0\}$ *except a finite Lebesgue measure set,*

$$NC1(H_{\rho_t}) \leq \delta_0.$$

The proof of Corollary 4.9 is deferred to Appendix C.3, and the result implies that $\liminf_{t \to +\infty} NC1(H_{\rho_t}) \leq \delta_0$. Corollary 4.9 implies that, the three-layer model (almost) always achieve NC1 solution, for long enough training, which explains the prevalence of neural collapse in practice.

**Imbalancedness after gradient-based training.** The numerical results of Figure 1 show that, even if the solution obtained via gradient descent is not balanced, its training loss and gradient norm are still small and, therefore, as predicted by our analysis, it satisfies NC1. As a normalized balancedness measure, we use

$$NB(\rho, W) = \frac{\|\mathbb{E}_\rho[aa^\top] - c_* W W^\top\|_{op}}{\|\mathbb{E}_\rho[aa^\top]\|_{op}}, \tag{25}$$

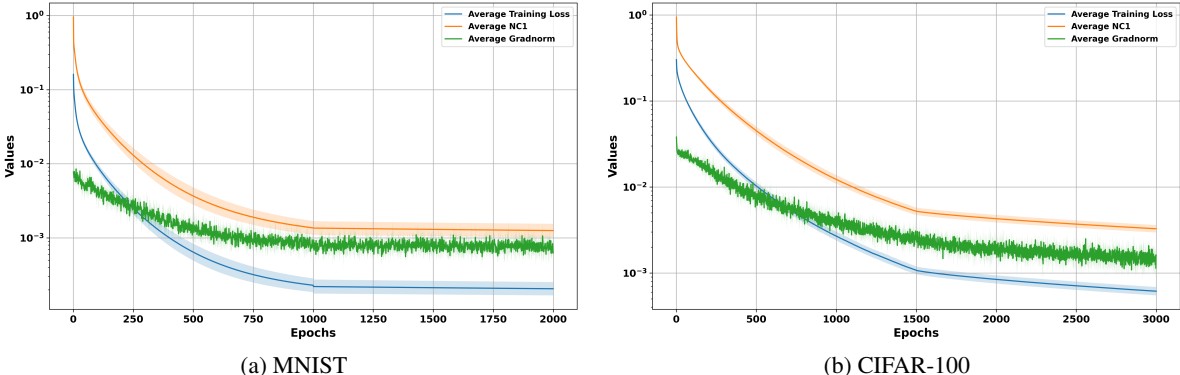

(a) MNIST                    (b) CIFAR-100

Figure 1: Average training loss (blue), NC1 (orange) and gradient norm (green) during SGD training. We report the average for 4 independent experiments, as well as the confidence interval at 1 standard deviation.

with

$$c_* = \arg\min_c \|\mathbb{E}_\rho[aa^\top] - cWW^\top\|_F^2. \quad (26)$$

This captures the extent to which $\mathbb{E}_\rho[aa^\top]$ and $WW^\top$ are proportional. We then train the three-layer neural network $f_N(x) = W^\top h_N(x)$, where the output $h_N(x)$ of the first two layers is given by (2). Specifically, we consider the following two settings.

*Setting (a): MNIST.* We relabel the dataset into $q = 3$ classes taking the original label modulo 3, and we randomly pick 10000 samples in each new class for training. The input dimension is $d = 784$, the number of neurons in the first layer is $N = 6272$, and the number of neurons in the second one is $p = 16$. We train the model with SGD of batch size 64 and learning rate $\eta \in \{0.001, 0.01\}$, using the smaller (larger) learning rate for the first (second) half of the epochs. We pick weight decay $\lambda_W = \lambda_\rho = 10^{-4}$, but add no noise ($\beta^{-1} = 0$) and fix the last linear layer at initialization, which produces an imbalanced network. In fact, at convergence, $NB(\rho, W) = \{0.8093, 0.7149, 0.5895, 0.8531\}$ in our 4 independent experiments. We also plot the evolution of the normalized balancedness metric in (25) as a function of the number of training epochs in Figure 2a of Appendix E, which shows that the network does not achieve balancedness throughout training. However, even if the network is not balanced, Figure 1a still shows that the NC1 metric decreases and flattens to a rather low value, following the same pattern as the training loss and the gradient norm.

*Setting (b): CIFAR-100.* We perform classification on super-classes using pretrained ResNet50 features. Specifically, we consider the 3 super-classes ["aquatic mammals", "large carnivores", "people"], with each super-class containing 5 original classes and 500 samples in total. We then take a ResNet50 pretrained on ImageNet-1K, extract the penultimate-layer features of the training set, and use such features as training data. The input dimension is $d = 2048$, the number of neurons in the first layer is $N = 16384$, and the number of neurons in the second one is $p = 64$. We train the model with noisy SGD of batch size 64, pick weight decay $\lambda_W = \lambda_\rho = \beta = 10^{-4}$ and learning rate $\eta \in \{0.001, 0.0001\}$, using the smaller (larger) learning rate for the first (second) half of the epochs. As in the previous case, the network does not achieve balancedness throughout training: at convergence, $NB(\rho, W) = \{0.5386, 0.2779, 0.4141, 0.5257\}$ in the 4 independent experiments; see also Figure 2b in Appendix E for a plot of the metric in (25) as a function of the number of training epochs. Nevertheless, the NC1 metric decreases with the loss and the gradient norm, reaching a small value at the end of training, see Figure 1b.

Similar results are reported for the training of ResNet-18 and VGG-11 in Appendix E. The implementation of the experiments is publicly available at the GitHub repository `https://github.com/DiyuanWu/icml25_expr`.

## 5. Within-class Variability Collapse and Generalization

While neural collapse is widely known as a phenomenon occurring at training time, it does not necessarily imply that the test error is small (Hui et al., 2022, Section 4). We now show that, for well-separated datasets, training via gradient flow implies both approximate NC1 *and* small test error.

**Problem setting.** We make the following additional assumptions (consistent with Assumption 1).

**Assumption 3.** We set $\gamma = 1$, and assume the activation function $\sigma$ to be the sigmoid function, i.e., $\sigma(z) = \frac{1}{1+e^{-z}}$. We further assume there are $q$ classes and $n$ data points $(x_j, y_j)$ sampled i.i.d. from $\mathcal{D}$, with $m$ points for each class and $x_j \sim \mathcal{D}(x_j|y_j)$. Each class is balanced, in the sense that $\int \mathcal{D}(\cdot, e_k) = 1/q$ for all $k$.

**Algorithm 1** Two-stage gradient flow

**Initialization:** Let $W_0 = [e_1, \ldots, e_q]^\top$.
**Stage 1:** Fix $W_0$ and obtain $\rho_1 = \arg\min_\rho \mathcal{E}_n^R(\rho, W_0)$.
**Stage 2:** Initialize at $(W_0, \rho_1)$ and run the following Wasserstein gradient flow:
$$\mathrm{d}W_t = -\nabla_W \mathcal{L}_{\lambda,n}(\rho_t, W_t)\mathrm{d}t;$$
$$\mathrm{d}\theta_t = -\nabla_\theta \frac{\delta}{\delta\rho}\mathcal{L}_{\lambda,n}(\rho_t, W_t)(\theta_t)\mathrm{d}t + \sqrt{2\beta^{-1}}\mathrm{d}B_t.$$

For technical reasons, we consider the approximated model obtained by truncating the second layer, i.e., $h_\rho^R(x) = \mathbb{E}_\rho[\tau_R(a)\sigma(u^\top x)]$, where $\tau_R : \mathbb{R} \to \mathbb{R}$ is a smooth function applied component-wise such that

$$\tau_R(z) = \begin{cases} z, & \text{for } |z| \leq R, \\ R + C_0, & \text{for } |z| \geq 2R, \\ \text{smooth interpolation}, & \text{for } R < |z| < 2R. \end{cases}$$

Notably, for large enough $R$, the derivative of $\tau$ satisfies

$$\tau_R'(z) = \begin{cases} 1, & \text{for } |z| \leq R, \\ 0, & \text{for } |z| \geq 2R, \\ \leq C_0, & \text{for } R < |z| < 2R. \end{cases}$$

We also denote $H_\rho^R = [h_\rho^R(x_1), \ldots, h_\rho^R(x_n)] \in \mathbb{R}^{p \times n}$, and define the loss and free energy w.r.t. the approximated second layer as

$$\mathcal{L}_n^R(\rho, W) = \frac{1}{2n}\|W^\top H_\rho^R - Y\|_F^2,$$

$$\mathcal{E}_n^R(\rho, W; \beta) = \mathcal{L}_n^R(\rho, W) + \frac{\beta^{-1}}{2}\|W\|_F^2$$
$$+ \frac{\beta^{-1}}{2}\mathbb{E}_\rho[\|\theta\|_2^2] + \beta^{-1}\mathbb{E}_\rho[\log\rho].$$

We remark that the technical reason for having an approximated second layer is to ensure the uniqueness of the global optimum for the Gibbs minimizer as discussed in Proposition 5.1. Nevertheless, our results in Section 5 hold uniformly for large enough $R$, which means that we also expect the same conclusion for the original model which corresponds to $R = +\infty$.

**Two stage training algorithm.** Our result holds for the two-stage training described in Algorithm 1. Specifically, in **Stage 1**, we aim to find the global optimum of $\mathcal{E}_n^R(W_0, \rho)$ having fixed $W_0$, and in Proposition 5.1 below we show that, for all fixed non-zero $W_0$, $\mathcal{E}_n^R(W_0, \rho)$ has a unique global minimizer in $\mathscr{P}_2(\mathbb{R}^{p+d})$. Furthermore, such global minimizer is achieved by noisy gradient flow as studied by (Suzuki et al., 2024a). In **Stage 2**, we run a gradient flow on the free energy, as we did in Section 4.2.

**Proposition 5.1.** *Under Assumption 1, for any fixed non-zero $W$, $\mathcal{E}_n^R(\rho, W)$ is strongly convex in $\rho$, and there exist a unique global minimizer $\rho$ with the following Gibbs form:*

$$\rho(\theta) \propto \exp\left(-\frac{\beta}{n}\tau_R(a)^\top W(W^\top H_\rho^R - Y)\sigma(X^\top u) - \frac{\beta}{2}\|\theta\|_2^2\right).$$
(27)

*Proof.* The result follows from the strong convexity of $\mathcal{E}_n^R(\rho, W)$. In fact, $\frac{1}{2n}\|W^\top H_\rho^R - Y\|_F^2$ is convex in $H_\rho^R$, $H_\rho^R$ is linear in $\rho$, the $L^2$-regularization is convex and the entropic regularization is strongly convex. Then, the claim is a consequence of (Hu et al., 2021, Proposition 2.5). □

**Test error analysis.** We start by introducing $(\tau, M)$-linearly separable data, which intuitively corresponds to each class being linearly separable w.r.t. the others.

**Definition 5.2.** We say that the data distribution $\mathcal{D}$ of a $q$-class classification problem is bounded and $(\tau, M)$-linearly separable if, for each $k$, there exist $\hat{u}_k$ s.t. $\|\hat{u}_k\|_2^2 \leq M^2$ and

$$\hat{u}_k^\top x \begin{cases} \geq \tau, & \text{if } x \in \text{supp}(\mathcal{D}(\cdot|e_k)), \\ < -\tau & \text{if } x \in \text{supp}(\mathcal{D}(\cdot|e_{k'})), \text{ for all } k' \neq k. \end{cases}$$

Given a predictor $f : \mathbb{R}^d \to \mathbb{R}^q$, we aim to bound the mismatch error:

$$\mathbf{err}_{\text{test}}(f; \mathcal{D}) = \frac{1}{q}\sum_{k=1}^q \Pr_{x \sim \mathcal{D}(\cdot|e_k)}[\text{One-Hot}(f(x)) \neq e_k],$$

where we define the function One-Hot : $\mathbb{R}^q \to \mathbb{R}^q$ as

$$[\text{One-Hot}(f)]_i = \begin{cases} 1, & \text{if } i = \arg\max_i[f]_i, \\ 0, & \text{else.} \end{cases}$$

While the $(\tau, M)$-linear separability of the data seem to be restrictive, we remark that in general it is not true that NC1 and vanishing test error co-occur under MSE loss without any assumptions on the data distribution. In fact, the occurrence of NC1 implies that the model overfits the data, and overfitting is not always benign without additional assumptions (Bartlett et al., 2021). In this sense, showing the co-occurrence of NC1 and vanishing test error may be regarded as a harder problem than benign overfitting.

We now show that training on a $(\tau, M)$-linearly separable dataset leads to both neural collapse and test error vanishing in the number of training samples $n$.

**Theorem 5.3.** *Under Assumptions 1 and 3, let the data distribution be bounded and $(\tau, M)$-linear separable as per Definition 5.2. Pick $R > 1$ large enough, $n$ large enough, and $\beta = \left(640C_1^2 nC_9^2\frac{1}{\delta_0}\right)^6$. Then, for any $(\rho_t, W_t)$ obtained by Stage 2 of Algorithm 1, we have*

$$\mathbf{err}_{test}(f(\cdot; \rho_t, W_t); \mathcal{D}) \leq C_{10}\log(C_{11}n/\delta_0)\sqrt{\frac{1}{2n}}$$
$$+ 6q\sqrt{\frac{\log(2/\delta)}{n}},$$

*with probability at least $1 - \delta$. Furthermore, there exists $T(\beta)$ s.t. for all $t > T(\beta)$ except a finite Lebesgue measure set,*

$$NC1(H_{\rho_t}) \leq \delta_0.$$

The constants $C_9, C_{10}, C_{11}$ depend on $d, p, C_0, C_1, M, \tau$, but not on $n$. Their expression is provided in Theorem D.6, whose statement and proof are in Appendix D.1. The argument uses Rademacher complexity bounds for neural networks in the mean-field regime as in (Chen et al., 2020; Suzuki et al., 2024b; Takakura & Suzuki, 2024), and the key component is to control the dependence of the constant $C_{10}$ on $n$. This is achieved by noting that, for a $(\tau, M)$-separated data distribution, a two-layer network with constant number of neurons approximately interpolates the data.

In a nutshell, Theorem 5.3 provides a sufficient condition on the data distribution to achieve both NC1 and vanishing test error. Although for simplicity in the statement we pick a specific value for $\beta$, we note that a similar result would hold for $\left(640C_1^2 nC_9^2 \frac{1}{\delta_0}\right)^6 \leq \beta \leq \mathcal{O}(\text{poly}(n))$.

## 6. Conclusions and Future Directions

In this work, we consider a three-layer neural network in the mean-field regime and give rather general sufficient conditions for within-class variability collapse (namely, NC1) to occur. We then show that *(i)* training the three-layer neural network with gradient flow satisfies these conditions, and *(ii)* a vanishing test error is compatible with neural collapse at training time. Taken together, our results connect representation geometry to loss landscape, gradient flow dynamics and generalization, offering new insights into gradient-based optimization in deep learning.

Three interesting future directions include: *(i)* establishing more general conditions (either necessary or sufficient) that guarantee both neural collapse during training and vanishing test error; *(ii)* tackling the challenging case in which there is a non-linearity between the last two layers – a setting where the properties of neural collapse have been proved in the UFM framework for binary classification (Súkeník et al., 2023); and *(iii)* extending the results to cross-entropy loss, which is more commonly used for classification. The main technical difficulty for the latter is to rule out the possibility that different data points in the same class could have different logits, which appears challenging even when the loss is small.

## Acknowledgements

This research was funded in whole or in part by the Austrian Science Fund (FWF) 10.55776/COE12. For the purpose of open access, the authors have applied a CC BY public copyright license to any Author Accepted Manuscript version arising from this submission. The authors would like to thank Peter Súkeník for general helpful discussions and for pointing out that all the stationary points are approximately proportional in the case without entropic regularization.

## Impact Statement

This paper presents work whose goal is to advance the field of Machine Learning. There are many potential societal consequences of our work, none which we feel must be specifically highlighted here.

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

## A. Technical Lemmas

**Lemma A.1** (Properties of Kronecker product and vectorization). *The following properties hold:*

1. $vec(ABC) = (C^\top \otimes A)vec(B)$.

2. $(A \otimes B)(C \otimes D) = (AC) \otimes (BD)$ *if one can form matrix product $AC$ and $BD$.*

3. *Singular space of Kronecker product: given two matrices $A \in \mathbb{R}^{m \times n}$ and $B \in \mathbb{R}^{p \times q}$, let their SVD be*

$$A = U_A S_A V_A^\top, \quad B = U_B S_B V_B^\top,$$

*where $U_A \in \mathbb{R}^{m \times m}, S_A \in \mathbb{R}^{m \times n}, V_A \in \mathbb{R}^{n \times n}$ and $U_B \in \mathbb{R}^{p \times p}, S_B \in \mathbb{R}^{p \times q}, V_B \in \mathbb{R}^{q \times q}$. Then, the SVD of $A \otimes B$ reads*

$$A \otimes B = (U_A \otimes U_B)(S_A \otimes S_B)(V_A \otimes V_B)^\top.$$

*Proof.* The first two claims can be easily verified. For the third, we have:

$$
\begin{aligned}
(U_A \otimes U_B)(S_A \otimes S_B)(V_A \otimes V_B)^\top &= (U_A \otimes U_B)(S_A \otimes S_B)(V_A^\top \otimes V_B^\top) \\
&= (U_A \otimes U_B)((S_A V_A^\top) \otimes (S_B V_B^\top)) \\
&= ((U_A S_A V_A^\top) \otimes (U_B S_B V_B^\top)) \\
&= A \otimes B,
\end{aligned}
$$

and

$$(U_A \otimes U_B)^\top (U_A \otimes U_B) = (U_A^\top U_A) \otimes (U_B^\top U_B) = I_m \otimes I_p = I_{mp},$$

which gives the desired result. $\square$

**Lemma A.2.** *Given two matrix $A, B \in \mathbb{R}^{n \times n}$, assume that $A$ is invertible and $A + B$ is invertible , then we have:*

$$(A + B)^{-1} = A^{-1} - (A + B)^{-1} B A^{-1}.$$

*Proof.* Let $(A + B)^{-1} = A^{-1} + C$ where we aim to compute $C$. Then, we have $(A + B)A^{-1} + (A + B)C = I$. This implies that $BA^{-1} + (A + B)C = 0$, and we have $C = -(A + B)^{-1}BA^{-1}$, which gives the desired result. $\square$

**Lemma A.3.** *Let $\rho(\theta) \in \mathscr{P}_2(\mathbb{R}^D)$ be an absolutely continuous measure, $\mathcal{L}(\rho) : \mathscr{P}_2 \to \mathbb{R}$ be a non-negative functional, and*

$$\mathcal{E}(\rho) = \mathcal{L}(\rho) + \frac{\lambda}{2}\mathbb{E}_\rho[\|\theta\|_2^2] + \beta^{-1}\mathbb{E}_\rho[\log \rho].$$

*Then, for any $\rho \in \mathscr{P}_2(\mathbb{R}^D)$, we have that*

$$
\begin{aligned}
\mathcal{L}(\rho) &\leq \mathcal{E}(\rho) + \beta^{-1}\frac{D}{2}\log\frac{2\pi}{\lambda\beta}, \\
D_{KL}(\rho\|\rho_0) &\leq \beta\left(\mathcal{E}(\rho) + \beta^{-1}\frac{D}{2}\log\frac{2\pi}{\lambda\beta}\right), \\
\mathbb{E}_\rho[\|\theta\|_2^2] &\leq 4\lambda^{-1}\mathcal{E}(\rho) + 4\lambda^{-1}\beta^{-1}\left(1 + D\log\frac{8\pi}{\lambda\beta}\right), \\
\mathcal{L}(\rho) + \frac{\lambda}{2}\mathbb{E}_\rho[\|\theta\|_2^2] &\leq 3\mathcal{E}(\rho) + \beta^{-1}\frac{D}{2}\log\frac{2\pi}{\lambda\beta} + 2\beta^{-1}\left(1 + D\log\frac{8\pi}{\lambda\beta}\right),
\end{aligned}
$$

*where $\rho_0 \propto \exp\left(-\beta\lambda\|\theta\|_2^2/2\right)$.*

*Proof.* Note that

$$\beta^{-1}\frac{\beta\lambda}{2}\mathbb{E}_\rho[\|\theta\|_2^2] + \beta^{-1}\mathbb{E}_\rho[\log\rho] = \beta^{-1}\mathbb{E}_\rho\left[\log\frac{\rho}{(2\pi/(\beta\lambda))^{-D/2}\exp(-\beta\lambda\|\theta\|_2^2/2)}\right] - \beta^{-1}\frac{D}{2}\log\frac{2\pi}{\beta\lambda}$$

$$\geq -\beta^{-1}\frac{D}{2}\log\frac{2\pi}{\lambda\beta},$$

where the last passage follows from the non-negativity of the KL divergence. This implies that

$$\mathcal{E}(\rho) = \mathcal{L}(\rho) + \beta^{-1}D_{KL}(\rho\|\rho_0) - \beta^{-1}\frac{D}{2}\log\frac{2\pi}{\lambda\beta} \geq \mathcal{L}(\rho) - \beta^{-1}\frac{D}{2}\log\frac{2\pi}{\lambda\beta},$$

which gives the first two inequalities. The third inequality on the second moment follows from (Mei et al., 2018, Equation 10.12), and the final equality comes from combining the first and third inequality. $\qquad\square$

## B. Proofs for Section 4.1

### B.1. Proof of Theorem 4.2

First, given $\rho, W$, we define

$$\Delta_a(\theta;\rho,W) := \nabla_a\frac{\delta}{\delta\rho}\mathcal{E}_n(\rho,W)(\theta) = \frac{\gamma}{n}W(\gamma W^\top H_\rho - Y)\sigma(X^\top u) + \lambda_\rho a + \beta^{-1}\nabla_a\log\rho(\theta), \quad \rho \text{ a.s.} \qquad (28)$$

and we will use the shorthand $\Delta_a$ in the rest of the proof. By the definition of $\epsilon_S$-stationary point, we have

$$\mathbb{E}_\rho[\|\Delta_a\|_2^2] \leq \epsilon_S^2.$$

By rearranging terms in (28), we have

$$a = -\frac{\gamma\lambda_\rho^{-1}}{n}W(\gamma W^\top H_\rho - Y)\sigma(X^\top u) - \lambda_\rho^{-1}\beta^{-1}\nabla_a\log\rho(\theta) + \lambda_\rho^{-1}\Delta_a, \quad \rho \text{ a.s.}$$

which implies that

$$H_\rho = \mathbb{E}_\rho[a\sigma(u^\top X)]$$

$$= -\lambda_\rho^{-1}\gamma W(\gamma W^\top H_\rho - Y)\frac{K_\rho(X,X)}{n} - \lambda_\rho^{-1}\beta^{-1}\mathbb{E}_\rho[\nabla_a\log\rho(\theta)\sigma(u^\top X)] + \lambda_\rho^{-1}\mathbb{E}_\rho[\Delta_a\sigma(u^\top X)].$$

We first show that the term $\mathbb{E}_\rho[\nabla_a\log\rho(\theta)\sigma(u^\top x)] = 0$, for any $x$. To see this, it is sufficient to show that

$$\int \partial_{a_1}\log\rho(\theta)\sigma(u^\top x)\,\rho(\mathrm{d}\theta) = 0.$$

Indeed, we have

$$\int \partial_{a_1}\log\rho(\theta)\sigma(u^\top x)\,\rho(\mathrm{d}\theta) = \int \partial_{a_1}\rho(\theta)\sigma(u^\top x)\,\mathrm{d}\theta = -\int \partial_{a_1}\sigma(u^\top x)\,\rho(\mathrm{d}\theta) = 0,$$

which implies that

$$H_\rho = -\lambda_\rho^{-1}\gamma W(\gamma W^\top H_\rho - Y)\frac{K_\rho(X,X)}{n} + \lambda_\rho^{-1}\mathbb{E}_\rho[\Delta_a\sigma(u^\top X)]. \qquad (29)$$

By multiplying both sides of (29) with $\gamma W^\top$ and subtracting $Y$, we get

$$\gamma W^\top H_\rho - Y = -\lambda_\rho^{-1}\gamma^2 W^\top W(\gamma W^\top H_\rho - Y)\frac{K_\rho(X,X)}{n} + \lambda_\rho^{-1}\gamma W^\top\mathbb{E}_\rho[\Delta_a\sigma(u^\top X)] - Y.$$

An application of the first property stated in Lemma A.1 gives that

$$vec(\gamma W^\top H_\rho - Y) = -\lambda_\rho \left( \gamma^2 \frac{K_\rho(X,X)}{n} \otimes (W^\top W) + \lambda_\rho I_{nq} \right)^{-1} \left( vec(Y) - \lambda_\rho^{-1} vec(\gamma W^\top \mathbb{E}_\rho[\Delta_a \sigma(u^\top X)]) \right). \quad (30)$$

Plugging the expression for $vec(\gamma W^\top H_\rho - Y)$ back to (29), we have

$$\begin{aligned}
vec(H_\rho) &= -\lambda_\rho^{-1} \left( \gamma \frac{K_\rho(X,X)}{n} \otimes W \right) vec(\gamma W^\top H_\rho - Y) + \lambda_\rho^{-1} vec(\mathbb{E}_\rho[\Delta_a \sigma(u^\top X)]) \\
&= \left( \gamma \frac{K_\rho(X,X)}{n} \otimes W \right) \left( \gamma^2 \frac{K_\rho(X,X)}{n} \otimes (W^\top W) + \lambda_\rho I_{nq} \right)^{-1} \left( vec(Y) - \lambda_\rho^{-1} vec(W^\top \mathbb{E}_\rho[\Delta_a \sigma(u^\top X)]) \right) \\
&\quad + \lambda_\rho^{-1} vec(\mathbb{E}_\rho[\Delta_a \sigma(u^\top X)]) \\
&= \left( \gamma \frac{K_\rho(X,X)}{n} \otimes W \right) \left( \gamma^2 \frac{K_\rho(X,X)}{n} \otimes (W^\top W) + \lambda_\rho I_{nq} \right)^{-1} vec(Y) + \boldsymbol{E}_1(\epsilon_S, \lambda_\rho; \gamma, W),
\end{aligned}$$

where we define the error vector as

$$\begin{aligned}
\boldsymbol{E}_1(\epsilon_S, \lambda_\rho; \gamma, W) &= -\lambda_\rho^{-1} \left( \gamma \frac{K_\rho(X,X)}{n} \otimes W \right) \left( \gamma^2 \frac{K_\rho(X,X)}{n} \otimes (W^\top W) + \lambda_\rho I_{nq} \right)^{-1} vec(\gamma W^\top \mathbb{E}_\rho[\Delta_a \sigma(u^\top X)]) \\
&\quad + \lambda_\rho^{-1} vec(\mathbb{E}_\rho[\Delta_a \sigma(u^\top X)]).
\end{aligned}$$

Now we aim to upper bound the error. To do so, we write

$$\begin{aligned}
&\|\boldsymbol{E}_1(\epsilon_S, \lambda_\rho; \gamma, W)\|_2^2 \\
&\leq 2\lambda_\rho^{-2} \left\| \left( \gamma \frac{K_\rho(X,X)}{n} \otimes W \right) \left( \gamma^2 \frac{K_\rho(X,X)}{n} \otimes (W^\top W) + \lambda_\rho I_{nq} \right)^{-1} vec(\gamma W^\top \mathbb{E}_\rho[\Delta_a \sigma(u^\top X)]) \right\|_2^2 \\
&\quad + 2\lambda_\rho^{-2} \|vec(\mathbb{E}_\rho[\Delta_a \sigma(u^\top X)])\|_2^2 \\
&\leq 2\lambda_\rho^{-2} \sigma_{\max} \left( \gamma \frac{K_\rho(X,X)}{n} \otimes W \right)^2 \sigma_{\max} \left( \left( \gamma^2 \frac{K_\rho(X,X)}{n} \otimes (W^\top W) + \lambda_\rho I_{nq} \right)^{-1} \right)^2 \|\gamma W^\top \mathbb{E}_\rho[\Delta_a \sigma(u^\top X)]\|_F^2 \\
&\quad + 2\lambda_\rho^{-2} \|\mathbb{E}_\rho[\Delta_a \sigma(u^\top X)]\|_F^2 \\
&\leq \left( 2\lambda_\rho^{-4} \gamma^4 C_1^2 \sigma_{\max}(W)^4 + 2\lambda_\rho^{-2} \right) \|\mathbb{E}_\rho[\Delta_a \sigma(u^\top X)]\|_F^2.
\end{aligned}$$
$$(31)$$

Then, we upper bound $\|\mathbb{E}_\rho[\Delta_a \sigma(u^\top X)]\|_F^2$ as follows:

$$\begin{aligned}
\|\mathbb{E}_\rho[\Delta_a \sigma(u^\top X)]\|_F^2 &\leq \mathbb{E}_\rho[\|\Delta_a \sigma(u^\top X)\|_F^2] \\
&= \mathbb{E}_\rho[\|\Delta_a\|_2^2 \|\sigma(u^\top X)\|_2^2] \\
&\leq C_1^2 n \mathbb{E}_\rho[\|\Delta_a\|_2^2] \\
&\leq C_1^2 n \epsilon_S^2.
\end{aligned} \quad (32)$$

Combining (31) and (32), (6)-(7) readily follow.

To obtain (9), we first show that, when $W^\top W$ is full rank, the following equality holds

$$\begin{aligned}
&\left( \gamma \frac{K_\rho(X,X)}{n} \otimes W \right) \left( \gamma^2 \frac{K_\rho(X,X)}{n} \otimes W^\top W + \lambda_\rho I_{nq} \right)^{-1} \\
&\quad = \gamma^{-1}(I_n \otimes (W(W^\top W)^{-1})) - \lambda_\rho \gamma^{-1}(I_n \otimes (W(W^\top W)^{-1})) \left( \frac{K_\rho(X,X)}{n} \otimes W^\top W + \lambda_\rho I_{nq} \right)^{-1}.
\end{aligned}$$
$$(33)$$

To do so, define the eigen-decomposition of $\frac{K_\rho(X,X)}{n}$ and SVD of $W$ as follows:

$$\frac{K_\rho(X,X)}{n} = U_K \Sigma_K U_K^\top, \quad \gamma W = U_W S_W V_W^\top,$$

where $U_K \in \mathbb{R}^{n \times n}, \Sigma_K \in \mathbb{R}^{n \times n}, U_W \in \mathbb{R}^{p \times p}, S_W \in \mathbb{R}^{p \times q}, V_W \in \mathbb{R}^{q \times q}$. By Lemma A.1, we have that

$$\gamma \frac{K_\rho(X,X)}{n} \otimes W = (U_K \otimes U_W)(\Sigma_K \otimes S_W)(U_K \otimes V_W)^\top, \quad \gamma^2 \frac{K_\rho(X,X)}{n} \otimes W^\top W = (U_K \otimes V_W)(\Sigma_K \otimes (S_W^\top S_W))(U_K \otimes V_W)^\top.$$

Thus, we have the following equalities:

$$\left( \gamma \frac{K_\rho(X,X)}{n} \otimes W \right) \left( \gamma^2 \frac{K_\rho(X,X)}{n} \otimes W^\top W + \lambda_\rho I_{nq} \right)^{-1}$$
$$= (U_K \otimes U_W)(\Sigma_K \otimes S_W)(U_K \otimes V_W)^\top (U_K \otimes V_W)(\Sigma_K \otimes (S_W^\top S_W) + \lambda_\rho I_{nq})^{-1}(U_K \otimes V_W)^\top$$
$$= (U_K \otimes U_W)(\Sigma_K \otimes S_W)(\Sigma_K \otimes (S_W^\top S_W) + \lambda_\rho I_{nq})^{-1}(U_K \otimes V_W)^\top.$$

For simplicity, we write the matrix $S_W = \begin{bmatrix} diag(\sigma_1, \ldots, \sigma_q) \\ 0_{p-q,q} \end{bmatrix}$ and define $S_W^{-1} = \begin{bmatrix} diag(\sigma_1^{-1}, \ldots, \sigma_q^{-1}) \\ 0_{p-q,q} \end{bmatrix}$. Here, given integers $n, m$, we define $0_{n,m}$ as the $n \times m$ matrix containing zeros. Clearly, we have that

$$S_W^{-1} S_W^\top = \begin{bmatrix} I_q & 0_{q,p-q} \\ 0_{p-q,q} & 0_{p-q,p-q} \end{bmatrix}.$$

Next, we observe that

$$(I_n \otimes S_W^{-1} S_W^\top)(\Sigma_K \otimes S_W) = \Sigma_K \otimes (S_W^{-1} S_W^\top S_W) = \Sigma_K \otimes S_W.$$

Thus, we can write

$$(\Sigma_K \otimes S_W)(\Sigma_K \otimes (S_W^\top S_W) + \lambda_\rho I_{nq})^{-1} = (I_n \otimes S_W^{-1} S_W^\top)(\Sigma_K \otimes S_W)(\Sigma_K \otimes (S_W^\top S_W) + \lambda_\rho I_{nq})^{-1}$$
$$= (I_n \otimes S_W^{-1})(I_n \otimes S_W^\top)(\Sigma_K \otimes S_W)(\Sigma_K \otimes (S_W^\top S_W) + \lambda_\rho I_{nq})^{-1}$$
$$= (I_n \otimes S_W^{-1})(\Sigma_K \otimes (S_W^\top S_W))(\Sigma_K \otimes (S_W^\top S_W) + \lambda_\rho I_{nq})^{-1}$$
$$= (I_n \otimes S_W^{-1}) \left( I - \lambda_\rho (\Sigma_K \otimes (S_W^\top S_W) + \lambda_\rho I_{nq})^{-1} \right),$$

which implies that

$$\left( \gamma \frac{K_\rho(X,X)}{n} \otimes W \right) \left( \gamma^2 \frac{K_\rho(X,X)}{n} \otimes W^\top W + \lambda_\rho I_{nq} \right)^{-1}$$
$$= (U_K \otimes U_W)(I_n \otimes S_W^{-1})(U_K \otimes V_W)^\top - \lambda_\rho (U_K \otimes U_W)(I_n \otimes S_W^{-1})(\Sigma_K \otimes (S_W^\top S_W) + \lambda_\rho I_{nq})^{-1}(U_K \otimes V_W)^\top.$$

Finally, we can verify that:

$$\gamma^{-1} I_n \otimes (W(W^\top W)^{-1}) = (U_K U_K^\top) \otimes (U_W S_W^{-1} V_W^\top)$$
$$= (U_K \otimes U_W)(I_n \otimes S_W^{-1})(U_K \otimes V_W)^\top,$$

and

$$\gamma^{-1}(I_n \otimes (W(W^\top W)^{-1})) \left( \gamma^2 \frac{K_\rho(X,X)}{n} \otimes W^\top W + \lambda_\rho I_{nq} \right)^{-1}$$
$$= (U_K \otimes U_W)(I_n \otimes S_W^{-1})(U_K \otimes V_W)^\top (U_K \otimes V_W)(\Sigma_K \otimes (S_W^\top S_W) + \lambda_\rho I_{nq})^{-1}(U_K \otimes V_W)^\top$$
$$= (U_K \otimes U_W)(I_n \otimes S_W^{-1})(\Sigma_K \otimes (S_W^\top S_W) + \lambda_\rho I_{nq})^{-1}(U_K \otimes V_W)^\top,$$

which gives (33).

From the above decomposition, we know that

$$\left(\gamma \frac{K_\rho(X,X)}{n} \otimes W\right)\left(\gamma^2 \frac{K_\rho(X,X)}{n} \otimes W^\top W + \lambda_\rho I_{nq}\right)^{-1} vec(Y)$$

$$=\gamma^{-1}(I_n \otimes (W(W^\top W)^{-1}))vec(Y) - \lambda_\rho \gamma^{-1}(I_n \otimes (W(W^\top W)^{-1}))\left(\gamma^2 \frac{K_\rho(X,X)}{n} \otimes W^\top W + \lambda_\rho I_{nq}\right)^{-1} vec(Y)$$

$$=\gamma^{-1}vec(W(W^\top W)^{-1}Y) - \lambda_\rho \gamma^{-1}(I_n \otimes (W(W^\top W)^{-1}))\left(\gamma^2 \frac{K_\rho(X,X)}{n} \otimes W^\top W + \lambda_\rho I_{nq}\right)^{-1} vec(Y),$$

where we use the first item of Lemma A.1 in the last passage. Let us now define

$$\widetilde{\boldsymbol{E}}_2(\epsilon_S, \lambda_\rho; \gamma, \rho, W) := -\lambda_\rho \gamma^{-1}(I_n \otimes (W(W^\top W)^{-1}))\left(\gamma^2 \frac{K_\rho(X,X)}{n} \otimes W^\top W + \lambda_\rho I_{nq}\right)^{-1} vec(Y)$$

$$=\gamma^{-1}(I_n \otimes (W(W^\top W)^{-1}))\left(vec(\gamma W^\top H_\rho - Y) - \left(\gamma^2 \frac{K_\rho(X,X)}{n} \otimes W^\top W + \lambda_\rho I_{nq}\right)^{-1} vec\left(\gamma W^\top \mathbb{E}_\rho[\Delta_a \sigma(u^\top X)]\right)\right),$$

where the second passage follows from (30). Using (6) (that we proved above), we have

$$vec(H_\rho) = vec(\gamma^{-1}W(W^\top W)^{-1}Y) + \widetilde{\boldsymbol{E}}_2(\epsilon_S, \lambda_\rho; \gamma, \rho, W) + \boldsymbol{E}_1(\epsilon_S, \lambda_\rho; \gamma, W).$$

It remains to upper bound $\|\widetilde{\boldsymbol{E}}_2(\epsilon_S, \lambda_\rho; \gamma, \rho, W)\|_2^2$. To this aim, we write

$$\|\widetilde{\boldsymbol{E}}_2(\epsilon_S, \lambda_\rho; \gamma, \rho, W)\|_2^2 \le 2\gamma^{-2}\sigma_{\max}(I_n \otimes (W(W^\top W)^{-1}))^2(\|\gamma W^\top H_\rho - Y\|_F^2 + \lambda_\rho^{-2}\gamma^2 \|W^\top \mathbb{E}_\rho[\Delta_a \sigma(u^\top X)]\|_F^2)$$

$$\le \frac{2n\gamma^{-2}\mathcal{L}_n(\rho, W) + 2\lambda_\rho^{-2}\|W^\top \mathbb{E}_\rho[\Delta_a \sigma(u^\top X)]\|_F^2}{\sigma_{\min}(W)^2}$$

$$\le \frac{2n\gamma^{-2}\mathcal{L}_n(\rho, W) + 2\lambda_\rho^{-2}\sigma_{\max}(W)^2\|\mathbb{E}_\rho[\Delta_a \sigma(u^\top X)]\|_F^2}{\sigma_{\min}(W)^2}.$$

Plugging in the bound in (32) gives

$$\|\widetilde{\boldsymbol{E}}_2(\epsilon_S, \lambda_\rho; \gamma, \rho, W)\|_2^2 \le \frac{2n\gamma^{-2}\mathcal{L}_n(\rho, W) + 2\lambda_\rho^{-2}\sigma_{\max}(W)^2 C_1^2 n\epsilon_S^2}{\sigma_{\min}(W)^2}. \tag{34}$$

By combining (7) and (34) with an application of the triangle inequality, the proof is complete.

### B.2. Proof of Lemma 4.3

*Proof of Lemma 4.3.* To upper bound $\sigma_{\max}(W)$, we directly use the definition in (4):

$$\sigma_{\max}(W)^2 \le \|W\|_F^2 \le 2\lambda_W^{-1}\mathcal{L}_{\lambda,n}(\rho, W) \le 2B(\lambda_W^0)^{-1}(\log \beta)^\alpha.$$

To lower bound $\sigma_{\min}(W)$, we start by showing that

$$\|H_\rho\|_F^2 \le \beta^{\epsilon_0}. \tag{35}$$

To see this, assume by contradiction that $\|H_\rho\|_F^2 > \beta^{\epsilon_0}$. Then, there exists $i$ such that $\|h_\rho(x_i)\|_2^2 > \frac{\beta^{\epsilon_0}}{n}$ and the following bounds hold:

$$\begin{aligned}
\|h_\rho(x_i)\|_2^2 &= \|\mathbb{E}_\rho[a\sigma(u^\top x_i)]\|_2^2 \\
&= \mathbb{E}_{\theta,\theta'}[a^\top a'\sigma(u^\top x_i)\sigma((u')^\top x_i)] \quad (\theta' \text{ is an independent copy of } \theta) \\
&\le \mathbb{E}_{\theta,\theta'}[|a|^\top |a'||\sigma(u^\top x_i)||\sigma((u')^\top x_i)|] \\
&\le C_1^2 \mathbb{E}_{\theta,\theta'}[|a|^\top |a|] \\
&\le C_1^2 \mathbb{E}_a[\|a\|_2]\mathbb{E}_{a'}[\|a\|_2] \\
&= C_1^2 \mathbb{E}_a[\|a\|_2]^2 \\
&\le C_1^2 \mathbb{E}_a[\|a\|_2^2],
\end{aligned}$$

which implies that $\mathbb{E}_\rho[\|a\|_2^2] > \frac{\beta^{\epsilon_0}}{C_1^2 n}$. By combining (15) with (4), we have

$$\mathbb{E}_\rho[\|a\|_2^2] \leq \mathbb{E}_\rho[\|\theta\|_2^2] \leq 2B(\lambda_\rho^0)^{-1}(\log \beta)^\alpha, \tag{36}$$

where we recall that $\lambda_\rho = \lambda_\rho^0 \beta^{-1}$. Note that, for all $\epsilon_0 \in (0, 1)$ and for all $\beta \geq e^{\frac{4\alpha}{\epsilon_0} \log \frac{2\alpha}{\epsilon_0}}$,

$$\frac{\beta^{\epsilon_0}}{(\log \beta)^\alpha} \geq \beta^{\epsilon_0/2}. \tag{37}$$

Thus, by taking

$$\beta \geq \max\{e^{\frac{4\alpha}{\epsilon_0} \log \frac{2\alpha}{\epsilon_0}}, (2C_1^2 nB(\lambda_\rho^0)^{-1})^{\frac{2}{\epsilon_0}}\}$$

as in (14), we have that $\frac{\beta^{\epsilon_0}}{C_1^2 n}$ is strictly larger than the RHS of (36), which is a contradiction.

Now, we are ready to argue that, for large enough $\beta$, $\sigma_{\min}(W) \geq \beta^{-\epsilon_0}$. Assume by contradiction that $\sigma_{\min}(W) < \beta^{-\epsilon_0}$, and w.l.o.g assume $[\Sigma_W]_{q,q} < \beta^{-\epsilon_0}$. Then, the following lower bound holds

$$\begin{aligned}
\mathcal{L}_{\lambda,n}(\rho, W) &\geq \mathcal{L}_n(\rho, W) \\
&= \frac{1}{2n}\|W^\top H_\rho - Y\|_F^2 \\
&= \frac{1}{2n}\|V_W \Sigma_W^\top U_W^\top H_\rho - Y\|_F^2 \\
&= \frac{1}{2n}\|\Sigma_W^\top U_W^\top H_\rho - V_W^\top Y\|_F^2 \\
&\geq \frac{1}{2n}\|[\Sigma_W]_{q,q}[U_W^\top H_\rho]_{q:} - [V_W^\top Y]_{q:}\|_2^2 \\
&\geq \frac{1}{4n}\|[V_W^\top Y]_{q:}\|_2^2 - \frac{1}{2n}\|[\Sigma_W]_{q,q}[U_W^\top H_\rho]_{q:}\|_2^2 \\
&= \frac{1}{4q} - \frac{1}{2n}\|[\Sigma_W]_{q,q}[U_W^\top H_\rho]_{q:}\|_2^2 \\
&> \frac{1}{4q} - \frac{1}{2n}\beta^{-\epsilon_0}.
\end{aligned} \tag{38}$$

Note that by taking $\beta \geq \left(\frac{4q}{n}\right)^{\frac{1}{\epsilon_0}}$, we have $\frac{1}{4q} - \frac{1}{2n}\beta^{-\epsilon_0} \geq \frac{1}{8q}$. Furthermore, for all $\beta \geq e^{4\alpha \log(2\alpha)}$, we have

$$\frac{\beta^{1/\alpha}}{\log \beta} > \beta^{1/(2\alpha)}.$$

Thus, by taking

$$\beta \geq \max\left\{e^{4\alpha \log(2\alpha)}, \left(\frac{4q}{n}\right)^{\frac{1}{\epsilon_0}}, 64(qB)^2\right\},$$

as in (14), we have that the RHS of (38) is strictly larger than $B\beta^{-1}(\log \beta)^\alpha$, which is a contradiction and concludes the proof.

$\square$

## B.3. Proof of Corollary 4.4

*Proof of Corollary 4.4.* From Theorem 4.2, we have

$$H_\rho = \gamma^{-1}W(W^\top W)^{-1}Y + \boldsymbol{E}_2.$$

Note that the NC1 metric is scale invariant, thus it is equivalent to compute $NC1(\gamma H_\rho)$. Later on, we will write $H_\rho := \gamma H_\rho$ with slight abuse of notation. Some manipulations give

$$\widetilde{H}_\rho = W(W^\top W)^{-1}\left(Y - \frac{1}{q}\mathbf{1}_q\mathbf{1}_n^\top\right) + \gamma \boldsymbol{E}_2\left(I_n - \frac{1}{n}\mathbf{1}_n\mathbf{1}_n^\top\right),$$

$$M_c = \frac{1}{m}W(W^\top W)^{-1}\left(Y - \frac{1}{q}\mathbf{1}_q\mathbf{1}_n^\top\right)Y^\top Y + \frac{\gamma}{m}\boldsymbol{E}_2\left(I_n - \frac{1}{n}\mathbf{1}_n\mathbf{1}_n^\top\right)Y^\top Y,$$

$$\widetilde{H}_\rho - M_c = \gamma\boldsymbol{E}_2\left(I_n - \frac{1}{n}\mathbf{1}_n\mathbf{1}_n^\top\right)\left(I_n - \frac{1}{m}Y^\top Y\right).$$

Thus, we have

$$\mathrm{Tr}\left\{(\widetilde{H}_\rho - M_c)^\top(\widetilde{H}_\rho - M_c)\right\} = \|\widetilde{H}_\rho - M_c\|_F^2$$

$$= \left\|\gamma\boldsymbol{E}_2\left(I_n - \frac{1}{n}\mathbf{1}_n\mathbf{1}_n^\top\right)\left(I_n - \frac{1}{m}Y^\top Y\right)\right\|_F^2$$

$$\leq 16\gamma^2\|\boldsymbol{E}_2\|_F^2,$$

$$\mathrm{Tr}\left\{\widetilde{H}_\rho^\top\widetilde{H}_\rho\right\} = \|\widetilde{H}_\rho\|_F^2$$

$$\geq \frac{1}{2}\left\|W(W^\top W)^{-1}\left(Y - \frac{1}{q}\mathbf{1}_q\mathbf{1}_n^\top\right)\right\|_F^2 - \|\gamma\boldsymbol{E}_2\left(I_n - \frac{1}{n}\mathbf{1}_n\mathbf{1}_n^\top\right)\|_F^2$$

$$\geq \frac{1}{2\sigma_{\max}^2(W)}\frac{(q-1)n}{q} - 4\gamma^2\|\boldsymbol{E}_2\|_F^2,$$

which concludes the proof. $\qquad\square$

### B.4. Proof of Lemma 4.5

*Proof of Lemma 4.5.* From the stationary condition, we obtain

$$\nabla_a\frac{\delta}{\delta\rho}\mathcal{E}_n(\rho, W) = \frac{\gamma}{n}W(\gamma W^\top H_\rho - Y)\sigma(X^\top u) + \lambda_\rho a + \beta^{-1}\nabla_a\log\rho(\theta) = 0, \quad \rho \ \text{a.s.}$$

$$\nabla_W\mathcal{E}_n(\rho, W) = \frac{\gamma}{n}H_\rho(\gamma W^\top H_\rho - Y)^\top + \lambda_W W = 0.$$

Rearranging the terms gives

$$a = -\frac{\lambda_\rho^{-1}\gamma}{n}W(\gamma W^\top H_\rho - Y)\sigma(X^\top u) - \beta^{-1}\lambda_\rho^{-1}\nabla_a\log\rho(\theta), \quad \rho \ \text{a.s.}$$

$$W = -\frac{\lambda_W^{-1}\gamma}{n}H_\rho(\gamma W^\top H_\rho - Y)^\top.$$

Then, we compute:

$$\mathbb{E}_\rho[aa^\top] = \mathbb{E}_\rho\left[-\frac{\lambda_\rho^{-1}\gamma}{n}W(\gamma W^\top H_\rho - Y)\sigma(X^\top u)a^\top - \beta^{-1}\lambda_\rho^{-1}(\nabla_a\log\rho(\theta))a^\top\right],$$

$$= -\frac{\lambda_\rho^{-1}\gamma}{n}W(\gamma W^\top H_\rho - Y)H_\rho^\top - \beta^{-1}\lambda_\rho^{-1}\mathbb{E}_\rho\left[(\nabla_a\log\rho(\theta))a^\top\right]$$

$$WW^\top = -\frac{\lambda_W^{-1}\gamma}{n}W(\gamma W^\top H_\rho - Y)H_\rho^\top.$$

Thus, $\lambda_W W W^\top - \lambda_\rho \mathbb{E}_\rho[aa^\top] = \beta^{-1}\mathbb{E}_\rho\left[(\nabla_a \log \rho(\theta))a^\top\right]$, and next we compute $\mathbb{E}_\rho\left[(\nabla_a \log \rho(\theta))a^\top\right]$. To do this, we note that:

$$
\begin{aligned}
\left[\mathbb{E}_\rho\left[(\nabla_a \log \rho(\theta))a^\top\right]\right]_{i,j} &= \mathbb{E}_\rho\left[\partial_{a_i} \log \rho(\theta)a_j\right] \\
&= \int \rho(\theta)\partial_{a_i} \log \rho(\theta)a_j \, \mathrm{d}\theta \\
&= \int a_j \partial_{a_i}\rho(\theta) \, \mathrm{d}\theta \\
&= -\int \rho(\theta)\partial_{a_i} a_j \, \mathrm{d}\theta \\
&= \begin{cases} -1, & i = j \\ 0, & i \neq j \end{cases}
\end{aligned}
$$

Thus, $\mathbb{E}_\rho\left[(\nabla_a \log \rho(\theta))a^\top\right] = -I_p$, which gives the desired result. $\qquad\square$

## C. Proofs in Section 4.2

### C.1. Proof of Lemma 4.6

*Proof of Lemma 4.6* . We note that $\mathcal{E}_n(\rho, W)$ is lower bounded and we have:

$$
\mathcal{E}_n(\rho_T, W_T) = \mathcal{E}_n(\rho_0, W_0) + \int_0^T \partial_t \mathcal{E}_n(\rho_t, W_t) \, \mathrm{d}t.
$$

A standard computation gives

$$
\partial_t \mathcal{E}_n(\rho_t, W_t) = -\mathbb{E}_{\rho_t}\left[\left\|\nabla_\theta \frac{\delta}{\delta\rho}\mathcal{E}_n(\rho_t, W_t)(\theta_t)\right\|_2^2\right] - \|\nabla_W \mathcal{L}_{\lambda,n}(\rho_t, W_t)\|_2^2,
$$

which implies that $\mathcal{E}_n(\rho_T, W_T)$ is a lower-bounded monotone decreasing sequence. Thus, $\lim_{T\to\infty} \mathcal{E}_n(\rho_T, W_T) = C < \infty$. The existence and boundedness of $\lim_{T\to\infty} \mathcal{E}_n(\rho_T, W_T)$ implies that, for any $\epsilon_S > 0$, there exists $T(\epsilon_S) > 0$ s.t. for all $t > T(\epsilon_S)$ except a finite Lebesgue measure set,

$$
\mathbb{E}_{\rho_t}\left[\left\|\nabla_\theta \frac{\delta}{\delta\rho}\mathcal{E}_n(\rho_t, W_t)(\theta_t)\right\|_2^2\right] + \|\nabla_W \mathcal{E}_n(\rho_t, W_t)\|_2^2 \leq \epsilon_S,
$$

which finishes the proof. $\qquad\square$

### C.2. Proof of Theorem 4.8.

**Theorem C.1** (Full statement of Theorem 4.8). *Let Assumptions 1, 2 hold, set $\lambda_\rho = \lambda_W = \beta^{-1}$ and*

$$
\gamma > C_3, \quad t_0 = \beta C_5.
$$

*Then, for any $\beta$ and any $t \geq t_0$, we have*

$$
\mathcal{L}_{\lambda,n}(\rho_t, W_t) \leq \beta^{-1}C_4,
$$

*where*

$$C_3 = \max\left\{\frac{4B_1}{R_W}, \frac{2\sqrt{2}B_3}{\sqrt{R_\rho}}\right\},$$

$$C_4 = 6\gamma^{-2}A_1^2 \max\left\{\frac{B_2^2}{B_1^2}, \frac{B_4^2}{B_3^2}\right\} + 6\gamma^{-2}A_2^2 + 3\frac{q(R_W+1)}{2} + 3R_\rho + \frac{p+d}{2}\log 2\pi + 2\left(1 + (p+d)\log 8\pi\right),$$

$$C_5 = \gamma^{-1}\min\left\{\frac{B_1}{B_2}, \frac{B_3}{B_4}\right\},$$

$$R_W = \frac{1}{2}, \quad R_\rho = \min\left\{p+d, \frac{\lambda_*^2}{64n^2C_1^2}\right\},$$

$$A_1 = \frac{\lambda_*}{2n}, \quad A_2 = 32\frac{\sqrt{2n(R_W+1)R_\rho}(2C_1 d\sqrt{p+d}+1)}{\lambda_*},$$

$$B_1 = \frac{2}{n}\sqrt{R_W+1}\sqrt{pn}(4C_1\sqrt{d(p+d)}+2C_1)2\sqrt{R_\rho}A_1^{-1},$$

$$B_2 = 2(R_W+1) + \frac{2}{n}\sqrt{R_W+1}\sqrt{pn}(4C_1\sqrt{d(p+d)}+2C_1)2\sqrt{R_\rho}A_2,$$

$$B_3 = 2p\sqrt{n}\sqrt{R_W+1}(4C_1\sqrt{d(p+d)}+2C_1)2\sqrt{R_\rho}(4C_1 d^{3/2}\sqrt{p+d}+2C_1 d)A_1^{-1},$$

$$B_4 = 2p\sqrt{n}\sqrt{R_W+1}(4C_1\sqrt{d(p+d)}+2C_1)2\sqrt{R_\rho}(4C_1 d^{3/2}\sqrt{p+d}+2C_1 d)A_2.$$

To prove the above Theorem C.1, we first define the following first hitting time for any fixed $R_W < 1, R_\rho < p+d$:

$$t_* = \min\{\inf\{t : \|W_t^\top W_t - W_0^\top W_0\|_{op} > R_W\}, \inf\{t : D_{KL}(\rho_t\|\rho_0) > R_\rho\}\}.$$

From the above definition of $t_*$, we have that, for $t \leq t_*$,

$$\|W_t^\top W_t - W_0^\top W_0\|_{op} \leq R_W, \qquad D_{KL}(\rho_t\|\rho_0) \leq R_\rho.$$

The next two lemmas (proved in Appendices C.2.1 and C.2.2) control the behavior of the dynamics before $t_*$.

**Lemma C.2.** *Let* $R_\rho \leq \min\{p+d, \frac{\lambda_*^2}{64n^2C_1^4}\}, R_W \leq \frac{1}{2}$. *For* $t \leq t_*$, *we have*

$$\sqrt{\mathcal{L}_n(\rho_t, W_t)} \leq \exp(-\gamma^2 A_1 t) + \gamma^{-1}\beta^{-1}A_2,$$

*where*

$$A_1 = \frac{\lambda_*}{4n}, \qquad A_2 = \frac{16\sqrt{2pn^2(R_W+1)R_\rho}(2C_1\sqrt{p+d}+1)}{\lambda_*}.$$

**Lemma C.3.** *Let* $R_\rho \leq \min\{p+d, \frac{\lambda_*^2}{64n^2C_1^4}\}, R_W \leq \frac{1}{2}$. *For* $t \leq t_*$, *we have*

$$\|W_t^\top W_t - W_0^\top W_0\|_{op} \leq \gamma^{-1}B_1 + \beta^{-1}B_2 t,$$

$$\sqrt{D_{KL}(\rho_t, \rho_0)} \leq \gamma^{-1}B_3 + \beta^{-1}B_4 t,$$

*where*

$$B_1 = \frac{4}{n}\sqrt{R_W+1}\sqrt{pn}(4C_1\sqrt{(p+d)}+2C_1)\sqrt{R_\rho}A_1^{-1},$$

$$B_2 = 2(R_W+1) + \frac{4}{n}\sqrt{R_W+1}\sqrt{pn}(4C_1\sqrt{(p+d)}+2C_1)\sqrt{R_\rho}A_2,$$

$$B_3 = \sqrt{2p}\sqrt{R_W+1}(8C_1\sqrt{p+d}+4C_1)A_1^{-1},$$

$$B_4 = \sqrt{2p}\sqrt{R_W+1}(8C_1\sqrt{p+d}+4C_1)A_2.$$

Now we are ready to prove the main theorem.

*Proof of Theorem C.1.* We pick $t_0 = \gamma^{-1}\beta \min\{\frac{B_1}{B_2}, \frac{B_3}{B_4}\}$, and we consider two cases. If $t_* < t_0$, then for any $t \leq t_* < t_0$, an application of Lemma C.3 gives

$$\|W_t^\top W_t - W_0^\top W_0\|_{op} \leq \gamma^{-1}B_1 + \beta^{-1}B_2 t \leq \gamma^{-1}B_1 + \beta^{-1}B_2 t_0 \leq 2\gamma^{-1}B_1,$$
$$D_{KL}(\rho_{t_*}\|\rho_0) \leq (\gamma^{-1}B_3 + \beta^{-1}B_4 t)^2 \leq (\gamma^{-1}B_3 + \beta^{-1}B_4 t_0)^2 \leq 4\gamma^{-2}B_3^2.$$

By picking

$$\gamma \geq \max\left\{\frac{4B_1}{R_W}, \frac{2\sqrt{2}B_3}{\sqrt{R_\rho}}\right\},$$

we get that:

$$\|W_t^\top W_t - W_0^\top W_0\|_{op} \leq \frac{R_W}{2}, \qquad D_{KL}(\rho_{t_*}\|\rho_0) \leq \frac{R_\rho}{2},$$

for all $t \leq t_0$ with $t_0 > t_*$, which contradicts the definition of $t_*$.

This implies that $t_* \geq t_0$ and, by Lemma C.2, we have

$$\mathcal{L}_n(\rho_{t_0}, W_{t_0}) \leq 2\exp\left(-2\gamma A_1 \min\left\{\frac{B_1}{B_2}, \frac{B_3}{B_4}\right\}\beta\right) + 2\gamma^{-2}\beta^{-2}A_2^2$$

$$\leq 2\left(\gamma A_1 \min\left\{\frac{B_1}{B_2}, \frac{B_3}{B_4}\right\}\beta\right)^{-2} + 2\gamma^{-2}\beta^{-2}A_2^2$$

$$\leq 2\gamma^{-2}\beta^{-2}A_1^2 \max\left\{\frac{B_2^2}{B_1^2}, \frac{B_4^2}{B_3^2}\right\} + 2\gamma^{-2}\beta^{-2}A_2^2.$$

We also have the following upper bound on $\|W_{t_0}\|_F^2$:

$$\|W_{t_0}\|_F^2 \leq q\|W_{t_0}^\top W_{t_0}\|_{op}$$
$$\leq q(\|W_{t_0}^\top W_{t_0} - W_0^\top W_0\|_{op} + \|W_0^\top W_0\|_{op})$$
$$\leq q(R_W + 1).$$

Thus, we can upper bound the free energy for all $t \geq t_0$ as

$$\mathcal{E}_n(\rho_t, W_t) \leq \mathcal{E}_n(\rho_{t_0}, W_{t_0}) \leq \mathcal{L}_n(\rho_{t_0}, W_{t_0}) + \frac{\beta^{-1}}{2}\|W_{t_0}\|_F^2 + \beta^{-1}D_{KL}(\rho_{t_0}\|\rho_0)$$

$$\leq 2\gamma^{-2}\beta^{-2}A_1^2 \max\left\{\frac{B_2^2}{B_1^2}, \frac{B_4^2}{B_3^2}\right\} + 2\gamma^{-2}\beta^{-2}A_2^2 + \frac{\beta^{-1}}{2}q(R_W + 1) + \beta^{-1}R_\rho.$$

Applying Lemma A.3 gives that for $t \geq t_0$:

$$\mathcal{L}_{\lambda,n}(\rho_t, W_t) \leq 3\mathcal{E}_n(\rho_t, W_t) + \beta^{-1}\frac{p+d}{2}\log 2\pi + 2\beta^{-1}(1 + (p+d)\log 8\pi) \leq \beta^{-1}C_4,$$

with

$$C_4 = 6\gamma^{-2}A_1^2 \max\left\{\frac{B_2^2}{B_1^2}, \frac{B_4^2}{B_3^2}\right\} + 6\gamma^{-2}A_2^2 + 3\frac{q(R_W + 1)}{2} + 3R_\rho + \frac{p+d}{2}\log 2\pi + 2(1 + (p+d)\log 8\pi),$$

where we use $\beta > 1$. This completes the proof. $\qquad\square$

### C.2.1. PROOF OF LEMMA C.2

We first compute the evolution of $\mathcal{L}_n(\rho_t, W_t) = \frac{1}{2n}\|\gamma W_t^\top H_{\rho_t} - Y\|_F^2$ under gradient flow:

$$\frac{\mathrm{d}}{\mathrm{d}t}\mathcal{L}_n(\rho_t, W_t) = \left\langle \frac{1}{n}r_t, \gamma\frac{\mathrm{d}}{\mathrm{d}t}W_t^\top H_{\rho_t}\right\rangle_F$$

$$= \gamma\left\langle \frac{1}{n}r_t, \left(\frac{\mathrm{d}}{\mathrm{d}t}W_t\right)^\top H_{\rho_t}\right\rangle_F + \gamma\left\langle \frac{1}{n}r_t, W_t^\top\left(\frac{\mathrm{d}}{\mathrm{d}t}H_{\rho_t}\right)\right\rangle_F,$$

where we define $r_t = \gamma W_t^\top H_{\rho_t} - Y \in \mathbb{R}^{q \times n}$.

The evolution of $W_t$ is computed as:

$$\frac{\mathrm{d}}{\mathrm{d}t} W_t = -\frac{\gamma}{n} H_{\rho_t} r_t^\top - \beta^{-1} W_t,$$

and the evolution of $H_{\rho_t}$ can be computed as:

$$\begin{aligned}
\frac{\mathrm{d}}{\mathrm{d}t} H_{\rho_t} &= \int a\sigma(u^\top X) \frac{\mathrm{d}}{\mathrm{d}t} \rho_t(\theta) \, \mathrm{d}\theta \\
&= \int a\sigma(u^\top X) \nabla_\theta \cdot (\rho_t(\theta) \nabla_\theta V_t(\theta)) \, \mathrm{d}\theta \\
&= -\int \nabla_a V_t(\theta) \sigma(u^\top X) + a(\nabla_u V_t(\theta))^\top X \mathrm{Diag}(\sigma'(u^\top X)) \, \rho_t(\mathrm{d}\theta),
\end{aligned}$$

where we define the potential $V_t(\cdot) : \mathbb{R}^{p+d} \to \mathbb{R}$ to be the first variation of the free energy

$$V_t(\theta) = \frac{\delta}{\delta\rho} \mathcal{E}_n(\rho_t, W_t)(\theta) = \left\langle \frac{1}{n} r_t, \gamma W_t^\top a\sigma(u^\top X) \right\rangle_F + \beta^{-1} \|\theta\|_2^2 + \beta^{-1} \log \rho_t(\theta),$$

and $\mathrm{Diag}(\sigma'(u^\top X)) \in \mathbb{R}^{n \times n}$ to be the diagonal matrix with $\sigma'(u^\top x_i)$ on the $i$-th diagonal entry. The gradient of the potential is given by

$$\begin{aligned}
\nabla_a V_t(\theta) &= \frac{\gamma}{n} W_t r_t \sigma(X^\top u) + \beta^{-1} a + \beta^{-1} \nabla_a \log \rho_t(\theta), \\
\nabla_u V_t(\theta) &= \frac{\gamma}{n} X \mathrm{Diag}(\sigma'(u^\top X)) r_t^\top W_t^\top a + \beta^{-1} u + \beta^{-1} \nabla_u \log \rho_t(\theta).
\end{aligned} \tag{39}$$

Thus, we can express the evolution of $H_{\rho_t}$ as follows:

$$\begin{aligned}
\frac{\mathrm{d}}{\mathrm{d}t} H_{\rho_t} = &-\frac{\gamma}{n} \int W_t r_t \sigma(X^\top u) \sigma(u^\top X) \, \rho_t(\mathrm{d}\theta) \\
&- \beta^{-1} \int (a + \nabla_a \log \rho_t(\theta)) \sigma(u^\top X) \, \rho_t(\mathrm{d}\theta) \\
&- \frac{\gamma}{n} \int aa^\top W_t r_t \mathrm{Diag}(\sigma'(u^\top X)) X^\top X \mathrm{Diag}(\sigma'(u^\top X)) \rho_t(\mathrm{d}\theta) \\
&- \beta^{-1} \int a(u + \nabla_u \log \rho_t(\theta))^\top X \mathrm{Diag}(\sigma'(u^\top X)) \, \rho_t(\mathrm{d}\theta).
\end{aligned}$$

Now, we can write the evolution of the empirical loss function as

$$\begin{aligned}
\frac{\mathrm{d}}{\mathrm{d}t} \mathcal{L}_n(\rho_t, W_t) = &-\frac{\gamma^2}{n^2} \langle r_t, r_t H_{\rho_t}^\top H_{\rho_t} \rangle_F - \frac{\gamma\beta^{-1}}{n} \langle r_t, W_t^\top H_{\rho_t} \rangle_F \\
&- \frac{\gamma^2}{n^2} \left\langle r_t, W_t^\top W_t r_t \int \sigma(X^\top u) \sigma(u^\top X) \, \rho_t(\mathrm{d}\theta) \right\rangle_F \\
&- \frac{\gamma\beta^{-1}}{n} \left\langle r_t, W_t^\top \int (a + \nabla_a \log \rho_t(\theta)) \sigma(u^\top X) \, \rho_t(\mathrm{d}\theta) \right\rangle_F \\
&- \frac{\gamma^2}{n^2} \left\langle r_t, \int W_t^\top aa^\top W_t r_t \mathrm{Diag}(\sigma'(u^\top X)) X^\top X \mathrm{Diag}(\sigma'(u^\top X)) \rho_t(\mathrm{d}\theta) \right\rangle_F \\
&- \frac{\gamma\beta^{-1}}{n} \left\langle r_t, W_t^\top \int a(u + \nabla_u \log \rho_t(\theta))^\top X \mathrm{Diag}(\sigma'(u^\top X)) \, \rho_t(\mathrm{d}\theta) \right\rangle_F.
\end{aligned}$$

We first control the potential positive terms via following lemma.

**Lemma C.4.** *Let $R_\rho \leq d + p$. Then, for $t \leq t_*$, we have*

$$\left| \langle r_t, W_t^\top H_{\rho_t} \rangle_F + \left\langle r_t, W_t^\top \int (a + \nabla_a \log \rho_t(\theta)) \sigma(u^\top X) \rho_t(\mathrm{d}\theta) \right\rangle_F \right.$$
$$\left. + \left\langle r_t, W_t^\top \int a(u + \nabla_u \log \rho_t(\theta))^\top X \mathrm{Diag}(\sigma'(u^\top X)) \rho_t(\mathrm{d}\theta) \right\rangle_F \right|$$
$$\leq 8\sqrt{2pn^2(R_W + 1)R_\rho}(2C_1\sqrt{p + d} + 1)\sqrt{\mathcal{L}_n(\rho_t, W_t)}.$$

*Proof.* We first note that

$$\int \nabla_a \log \rho_t(\theta) \sigma(u^\top x_j) \rho_t(\mathrm{d}\theta) = -\int \nabla_a(\sigma(u^\top x_j)) \rho_t(\mathrm{d}\theta) = 0,$$

$$\left[ \int a(\nabla_u \log \rho_t(\theta))^\top X \mathrm{Diag}(\sigma'(u^\top X)) \rho_t(\mathrm{d}\theta) \right]_{i,j} = \int a_i (\nabla_u \log \rho_t(\theta))^\top x_j \sigma'(u^\top x_j) \rho_t(\mathrm{d}\theta)$$
$$= -\int \rho_t(\theta) \nabla_u \cdot (a_i \sigma'(u^\top x_j) x_j) \, \mathrm{d}\theta \tag{40}$$
$$= -\int a_i \sigma''(u^\top x_j) \|x_j\|_2^2 \rho_t(\mathrm{d}\theta).$$

For simplicity, we define the function

$$g_{i,j}(\theta) = 2a_i\sigma(u^\top x_j) + a_i u^\top x_j \sigma'(u^\top x_j) - a_i \sigma''(u^\top x_j) \|x_j\|_2^2,$$
$$[G(\theta)]_{i,j} = g_{i,j}(\theta) \in \mathbb{R}^{p \times n}.$$

Then, we have

$$\left| \langle r_t, W_t^\top H_{\rho_t} \rangle_F + \left\langle r_t, W_t^\top \int (a + \nabla_a \log \rho_t(\theta)) \sigma(u^\top X) \rho_t(\mathrm{d}\theta) \right\rangle_F \right.$$
$$\left. + \left\langle r_t, W_t^\top \int a(u + \nabla_u \log \rho_t(\theta))^\top X \mathrm{Diag}(\sigma'(u^\top X)) \rho_t(\mathrm{d}\theta) \right\rangle_F \right|$$
$$= \left| \left\langle r_t, W_t^\top \int G(\theta) \rho_t(\mathrm{d}\theta) \right\rangle_F \right|$$
$$\leq \|r_t\|_F \|W_t\|_{op} \left\| \int G(\theta) \rho_t(\mathrm{d}\theta) \right\|_F$$
$$\leq \sqrt{2n\mathcal{L}_n(\rho_t, W_t)} \|W_t\|_{op} \sqrt{\sum_{i,j} \left( \int g_{i,j}(\theta) \rho_t(\mathrm{d}\theta) \right)^2}$$
$$\leq \sqrt{2n\mathcal{L}_n(\rho_t, W_t)} \|W_t\|_{op} \sqrt{\sum_{i,j} \left( \int g_{i,j}(\theta) (\rho_t - \rho_0)(\mathrm{d}\theta) \right)^2},$$

where in the last step we use that

$$\mathbb{E}_{\rho_0}[2a_i\sigma(u^\top x_j) + a_i u^\top x_j \sigma'(u^\top x_j) - a_i \sigma''(u^\top x_j) \|x_j\|_2^2]$$
$$= \mathbb{E}_{\rho_0}[a_i] \mathbb{E}_{\rho_0}[2\sigma(u^\top x_j) + u^\top x_j \sigma'(u^\top x_j) - \sigma''(u^\top x_j) \|x_j\|_2^2] = 0,$$

since $\rho_0 = \mathcal{N}(0, I_{p+d})$.

Following the computations in (Chen et al., 2020, Lemma A.1, Equation C.4 and C.5) and using Assumption 1, we have

$$\|\nabla_\theta g_{i,j}(\theta)\|_2 \leq 4C_1(\|\theta\|_2 + 1).$$

Thus, by (Chen et al., 2020, Lemma B.2), we obtain

$$\left| \int g_{i,j}(\theta)\,(\rho_t - \rho_0)(\mathrm{d}\theta) \right| \le (8C_1\sqrt{p+d}+4)\mathcal{W}_2(\rho_t, \rho_0).$$

Hence, we conclude that

$$\left| \left\langle r_t, W_t^\top \int G(\theta)\,\rho_t(\mathrm{d}\theta) \right\rangle \right| \le 8\sqrt{2pn^2(R_W+1)R_\rho}(2C_1\sqrt{p+d}+1)\sqrt{\mathcal{L}_n(\rho_t, W_t)},$$

where we use that, for $t < t_*$,

$$\mathcal{W}_2(\rho_t, \rho_0) \le 2\sqrt{D_{KL}(\rho_t||\rho_0)} \le 2\sqrt{R_\rho}, \quad \text{(by Talagrand's inequality, see (Chen et al., 2020, Lemma 5.4) )}$$

$$\|W_t\|_{op}^2 = \|W_t^\top W_t\|_{op} \le \|W_0^\top W_0\|_{op} + \|W_t^\top W_t - W_0^\top W_0\|_{op} \le R_W + 1.$$

This concludes the argument. $\qquad\square$

Next, we lower bound the negative terms. We first recall the definition the kernel:

$$K_\rho(X, X) = \int \sigma(X^\top u)\sigma(u^\top X)\,\rho(\mathrm{d}\theta).$$

Furthermore, by Lemma 4.7, $\lambda_{\min}(K_{\rho_0}(X, X)) \ge \lambda_* > 0$. As $\lambda_{\min}(W_0^\top W_0) = 1$, this implies that $\lambda_{\min}(K_{\rho_0}(X, X) \otimes W_0^\top W_0) \ge \lambda_*$. We then have the following lower bound at time $t < t_*$.

**Lemma C.5.** *Let $R_\rho \le \min\{p+d, \frac{\lambda_*^2}{64n^2 C_1^4}\}$ and $R_W \le \frac{1}{2}$. Then, for $t \le t_*$, we have*

$$\lambda_{\min}(K_{\rho_t}(X, X) \otimes (W_t^\top W_t)) \ge \frac{\lambda_*}{4}.$$

*Proof.* First, by Weyl's inequality we have

$$\lambda_{\min}(W_t^\top W_t) \ge \lambda_{\min}(W_0^\top W_0) - \|W_0^\top W_0 - W_t^\top W_t\|_{op} \ge 1 - R_W.$$

It remains to lower bound $\lambda_{\min}(K_{\rho_t}(X, X))$. To do so, note that

$$\begin{aligned}
|K_{\rho_t}(x_i, x_j) - K_{\rho_0}(x_i, x_j)| &= |\mathbb{E}_{\rho_t}[\sigma(u^\top x_i)\sigma(u^\top x_j)] - \mathbb{E}_{\rho_0}[\sigma(u^\top x_i)\sigma(u^\top x_j)]| \\
&\le 2C_1^2\mathcal{W}_1(\rho_t, \rho_0) \\
&\le 4C_1^2\sqrt{D_{KL}(\rho_t||\rho_0)} \le 4C_1^2\sqrt{R_\rho},
\end{aligned}$$

where in the first inequality we use Kantorovich-Rubinstein duality. Thus, we have

$$\|K_{\rho_t}(X, X) - K_{\rho_0}(X, X)\|_{op} \le \|K_{\rho_t}(X, X) - K_{\rho_0}(X, X)\|_F \le 4nC_1^2\sqrt{R_\rho},$$

which implies that

$$\lambda_{\min}(K_{\rho_t}(X, X)) \ge \lambda_{\min}(K_{\rho_0}(X, X)) - \|(K_{\rho_0}(X, X) - K_{\rho_t}(X, X)\|_{op} \ge \lambda_* - 4nC_1^2\sqrt{R_\rho}.$$

By picking $R_\rho \le \min\{p+d, \frac{\lambda_*^2}{64n^2 C_1^4}\}$ and $R_W \le \frac{1}{2}$, the claim follows:

$$\lambda_{\min}(K_{\rho_t}(X, X) \otimes (W_t^\top W_t)) = \lambda_{\min}(K_{\rho_t}(X, X))\lambda_{\min}(W_t^\top W_t) \ge \frac{\lambda_*}{4}.$$

$\qquad\square$

By combining the results of Lemmas C.4 and C.5, we have that, for $R_\rho \leq \min\{p+d, \frac{\lambda_*^2}{64n^2 C_1^4}\}$, $R_W \leq \frac{1}{2}$ and $t \leq t_*$,

$$
\begin{aligned}
\frac{\mathrm{d}}{\mathrm{d}t} \mathcal{L}_n(\rho_t, W_t) &\leq -\frac{\gamma^2}{n^2} \lambda_{\min}(K_{\rho_t}(X, X) \otimes (W_t^\top W_t)) \|r_t\|_F^2 + \frac{\gamma \beta^{-1}}{n} \Big| \langle r_t, W_t^\top H_{\rho_t} \rangle_F \\
&\quad + \left\langle r_t, W_t^\top \int (a + \nabla_a \log \rho_t(\theta)) \sigma(u^\top X) \, \rho_t(\mathrm{d}\theta) \right\rangle_F \\
&\quad + \left\langle r_t, W_t^\top \int a(u + \nabla_u \log \rho_t(\theta))^\top X \mathrm{Diag}(\sigma'(u^\top X)) \, \rho_t(\mathrm{d}\theta) \right\rangle_F \Big| \\
&\leq -\frac{\gamma^2}{n^2} \frac{\lambda_*}{4} 2n \mathcal{L}(\rho_t, W_t) + \frac{\gamma \beta^{-1}}{n} 8 \sqrt{2pn^2(R_W + 1)R_\rho} (2C_1 \sqrt{p+d} + 1) \sqrt{\mathcal{L}_n(\rho_t, W_t)}.
\end{aligned}
$$

By dividing both sides by $2\sqrt{\mathcal{L}_n(\rho_t, W_t)}$ and defining

$$
Z_1 = \frac{\lambda_*}{4n}, \qquad Z_2 = \frac{4\sqrt{2pn^2(R_W + 1)R_\rho}(2C_1\sqrt{p+d}+1)}{n},
$$

we get

$$
\frac{1}{2\sqrt{\mathcal{L}_n(\rho_t, W_t)}} \frac{\mathrm{d}}{\mathrm{d}t} \mathcal{L}_n(\rho_t, W_t) \leq -\gamma^2 Z_1 \sqrt{\mathcal{L}_n(\rho_t, W_t)} + \gamma \beta^{-1} Z_2.
$$

Note that $\frac{1}{2\sqrt{\mathcal{L}_n(\rho_t, W_t)}} \frac{\mathrm{d}}{\mathrm{d}t} \mathcal{L}_n(\rho_t, W_t) = \frac{\mathrm{d}}{\mathrm{d}t} \sqrt{\mathcal{L}_n(\rho_t, W_t)}$. Hence, an application of Gronwall's Lemma gives

$$
\begin{aligned}
\sqrt{\mathcal{L}_n(\rho_t, W_t)} &\leq \exp\left(-\gamma^2 Z_1 t\right) \left( \sqrt{\mathcal{L}_n(\rho_0, W_0)} - \gamma^{-1} \beta^{-1} \frac{Z_2}{Z_1} \right) + \gamma^{-1} \beta^{-1} \frac{Z_2}{Z_1} \\
&\leq \exp\left(-\gamma^2 Z_1 t\right) + \gamma^{-1} \beta^{-1} \frac{Z_2}{Z_1},
\end{aligned}
$$

which gives the desired result.

### C.2.2. PROOF OF LEMMA C.3

We first control the term $\|W_t^\top W_t - W_0^\top W_0\|_{op}$. Note that, as $W_t^\top W_t - W_0^\top W_0$ is symmetric, we have

$$
\|W_t^\top W_t - W_0^\top W_0\|_{op} = \max\{\lambda_{\max}(W_t^\top W_t - W_0^\top W_0), \lambda_{\max}(W_0^\top W_0 - W_t^\top W_t)\}.
$$

Then, for any fixed $v \in \mathbb{S}^{q-1}$, we have

$$
\begin{aligned}
\frac{\mathrm{d}}{\mathrm{d}t} v^\top (W_t^\top W_t - W_0^\top W_0) v &= \frac{\mathrm{d}}{\mathrm{d}t} v^\top W_t^\top W_t v \\
&= v^\top \left( \left(\frac{\mathrm{d}}{\mathrm{d}t} W_t\right)^\top W_t + W_t^\top \left(\frac{\mathrm{d}}{\mathrm{d}t} W_t\right) \right) v \\
&= -2\beta^{-1} v^\top W_t^\top W_t v - \frac{\gamma}{n} v^\top (r_t H_{\rho_t}^\top W_t + W_t^\top H_{\rho_t} r_t^\top) v \\
&\leq \frac{\gamma}{n} |v^\top (r_t H_{\rho_t}^\top W_t + W_t^\top H_{\rho_t} r_t^\top) v| \\
&\leq \frac{2\gamma}{n} \|r_t^\top v\|_2 \|H_{\rho_t}^\top W_t v\|_2 \\
&\leq \frac{2\gamma}{n} \|r_t^\top\|_F \|H_{\rho_t}^\top W_t\|_F \\
&= \frac{2\gamma}{n} \sqrt{2n\mathcal{L}_n(\rho_t, W_t)} \|H_{\rho_t}^\top W_t\|_F.
\end{aligned}
$$

To upper bound $\|H_{\rho_t}^\top W_t\|_F$, by using the same techniques in Lemma C.2, we have:

$$
\|H_{\rho_t}^\top W_t\|_F \leq \|W_t\|_{op} \|H_{\rho_t} - H_{\rho_0}\|_F,
$$

which, as $\|W_t\|_{op} \leq \sqrt{R_W + 1}$ and $\|\nabla_\theta a_i \sigma(u^\top x_j)\|_2 \leq 2C_1(\|\theta\|_2 + 1), \forall i \in [p], j \in [n]$, gives

$$
\begin{aligned}
\|H_{\rho_t}^\top W_t\|_F &\leq \sqrt{R_W + 1}\sqrt{pn}(4C_1\sqrt{p+d} + 2C_1)\mathcal{W}_2(\rho_t, \rho_0) \\
&\leq \sqrt{R_W + 1}\sqrt{pn}(4C_1\sqrt{p+d} + 2C_1)2\sqrt{R_\rho}.
\end{aligned}
\tag{41}
$$

Similarly, we have

$$
\begin{aligned}
\frac{\mathrm{d}}{\mathrm{d}t}v^\top(W_0^\top W_0 - W_t^\top W_t)v &= \frac{\mathrm{d}}{\mathrm{d}t} - v^\top W_t^\top W_t v \\
&= -v^\top \left(\left(\frac{\mathrm{d}}{\mathrm{d}t}W_t\right)^\top W_t + W_t^\top \left(\frac{\mathrm{d}}{\mathrm{d}t}W_t\right)\right) v \\
&= 2\beta^{-1}v^\top W_t^\top W_t v + \frac{\gamma}{n}v^\top(r_t H_{\rho_t}^\top W_t + W_t^\top H_{\rho_t} r_t^\top)v \\
&\leq 2\beta^{-1}\|W_t^\top W_t\|_{op} + \frac{2\gamma}{n}\sqrt{2n\mathcal{L}_n(\rho_t, W_t)}\|H_{\rho_t}^\top W_t\|_F \\
&\leq 2\beta^{-1}(R_W + 1) + \frac{2\gamma}{n}\sqrt{2n\mathcal{L}_n(\rho_t, W_t)}\|H_{\rho_t}^\top W_t\|_F.
\end{aligned}
$$

Thus, for any $t < t_*$ and any $v \in \mathbb{S}^{q-1}$, we have

$$
|v^\top(W_t^\top W_t - W_0^\top W_0)v| \leq 2\beta^{-1}(R_W + 1)t + \frac{4\gamma}{n}\sqrt{R_W + 1}\sqrt{pn}(4C_1\sqrt{(p+d)} + 2C_1)\sqrt{R_\rho}\int_0^t \sqrt{\mathcal{L}_n(\rho_s, W_s)}\,\mathrm{d}s,
$$

which implies

$$
\|W_t^\top W_t - W_0^\top W_0\|_{op} \leq 2\beta^{-1}(R_W + 1)t + \frac{4\gamma}{n}\sqrt{R_W + 1}\sqrt{pn}(4C_1\sqrt{(p+d)} + 2C_1)\sqrt{R_\rho}\int_0^t \sqrt{\mathcal{L}_n(\rho_s, W_s)}\,\mathrm{d}s.
$$

For simplicity, let us define

$$
Z_3 = 2(R_W + 1), \qquad Z_4 = \frac{4}{n}\sqrt{R_W + 1}\sqrt{pn}(4C_1\sqrt{(p+d)} + 2C_1)\sqrt{R_\rho}.
$$

By plugging in the result of Lemma C.2, we have:

$$
\begin{aligned}
\|W_t^\top W_t - W_0^\top W_0\|_{op} &\leq \beta^{-1}Z_3 t + \gamma Z_4 \int_0^t \left(\exp(-\gamma^2 A_1 s) + \gamma^{-1}\beta^{-1}A_2\right)\mathrm{d}s \\
&= \beta^{-1}(Z_3 + Z_4 A_2)t - \gamma Z_4 \gamma^{-2}A_1^{-1}\exp(-\gamma^2 A_1 s)|_{s=0}^t \\
&\leq \beta^{-1}(Z_3 + Z_4 A_2)t + \gamma^{-1}Z_4 A_1^{-1}.
\end{aligned}
$$

Next, we control the term $D_{KL}(\rho_t||\rho_0)$. First, the time derivative of $D_{KL}(\rho_t||\rho_0)$ is computed as follows:

$$
\begin{aligned}
\frac{\mathrm{d}}{\mathrm{d}t}D_{KL}(\rho_t||\rho_0) &= \frac{\mathrm{d}}{\mathrm{d}t}\left(\frac{1}{2}\mathbb{E}_{\rho_t}[\|\theta\|_2^2] + \mathbb{E}_{\rho_t}[\log \rho_t]\right) \\
&= \int \left(\frac{1}{2}\|\theta\|_2^2 + \log \rho_t(\theta)\right)\frac{\mathrm{d}}{\mathrm{d}t}\rho_t(\theta)\,\mathrm{d}\theta \\
&= \int \left(\frac{1}{2}\|\theta\|_2^2 + \log \rho_t(\theta)\right)\nabla_\theta \cdot (\rho_t(\theta)\nabla_\theta V_t(\theta))\,\mathrm{d}\theta \\
&= -\int \langle\theta + \nabla_\theta \log \rho_t(\theta), \nabla_\theta V_t(\theta)\rangle\rho_t(\mathrm{d}\theta).
\end{aligned}
$$

By recalling (39), we have

$$
\begin{aligned}
\frac{\mathrm{d}}{\mathrm{d}t} D_{KL}(\rho_t \| \rho_0) = & -\beta^{-1} \int \|\theta + \nabla_\theta \log \rho_t(\theta)\|_2^2 \, \rho_t(\mathrm{d}\theta) \\
& - \frac{\gamma}{n} \int (a + \nabla_a \log \rho_t(\theta))^\top W_t r_t \sigma(X^\top u) \, \rho_t(\mathrm{d}\theta) \\
& - \frac{\gamma}{n} \int (u + \nabla_u \log \rho_t(\theta))^\top X \mathrm{Diag}(\sigma'(u^\top X)) r_t^\top W_t^\top a \, \rho_t(\mathrm{d}\theta) \\
\leq & -\frac{\gamma}{n} \left\langle r_t, W_t^\top \int (a + \nabla_a \log \rho_t(\theta)) \sigma(u^\top X) \, \rho_t(\mathrm{d}\theta) \right\rangle_F \\
& - \frac{\gamma}{n} \left\langle r_t, W_t^\top \int a(u + \nabla_u \log \rho_t(\theta))^\top X \mathrm{Diag}(\sigma'(u^\top X)) \, \rho_t(\mathrm{d}\theta) \right\rangle_F \\
\leq & \left| \left\langle r_t, W_t^\top \int G(\theta) \, \rho_t(\mathrm{d}\theta) \right\rangle_F \right|.
\end{aligned}
$$

where by recalling (40) and defining

$$
\begin{aligned}
& g_{i,j}(\theta) = a_i \sigma(u^\top x_j) + a_i u^\top x_j \sigma'(u^\top x_j) + a_i \sigma''(u^\top x_j) \|x_j\|_2^2, \\
& [G(\theta)]_{i,j} = g_{i,j}(\theta) \in \mathbb{R}^{p \times n},
\end{aligned}
$$

we have

$$
\begin{aligned}
\frac{\mathrm{d}}{\mathrm{d}t} D_{KL}(\rho_t \| \rho_0) \leq & \frac{\gamma}{n} \int \langle r_t, W_t^\top G(\theta) \rangle \, \rho_t(\mathrm{d}\theta) \\
\leq & \frac{\gamma}{n} \|W_t\|_{op} \sqrt{2n \mathcal{L}_n(\rho_t, W_t)} \sqrt{\sum_{i,j} \left( \int g_{i,j}(\theta)(\rho_t - \rho_0)(\mathrm{d}\theta) \right)^2}.
\end{aligned}
$$

Following the computations in (Chen et al., 2020, Lemma A.1, Equation C.4 and C.5) and using Assumption 1, we have

$$
\|\nabla_\theta g_{i,j}(\theta)\|_2 \leq 4C_1(\|\theta\|_2 + 1).
$$

Thus, by (Chen et al., 2020, Lemma B.2), we obtain

$$
\left| \int g_{i,j}(\theta)(\rho_t - \rho_0)(\mathrm{d}\theta) \right| \leq (8C_1 \sqrt{p+d} + 4C_1) \mathcal{W}_2(\rho_t, \rho_0) \leq (8C_1 \sqrt{p+d} + 4C_1) 2 \sqrt{D_{KL}(\rho_t \| \rho_0)}.
$$

Hence, we conclude that

$$
\frac{\mathrm{d}}{\mathrm{d}t} D_{KL}(\rho_t \| \rho_0) \leq 2\gamma Z_5 \sqrt{D_{KL}(\rho_t \| \rho_0)} \sqrt{\mathcal{L}_n(\rho_t, W_t)},
$$

with

$$
Z_5 = \sqrt{2p} \sqrt{R_W + 1} (8C_1 \sqrt{p+d} + 4C_1).
$$

Thus, we have

$$
\frac{\mathrm{d}}{\mathrm{d}t} \sqrt{D_{KL}(\rho_t \| \rho_0)} \leq \gamma Z_5 \sqrt{\mathcal{L}_n(\rho_t, W_t)},
$$

which implies

$$
\begin{aligned}
\sqrt{D_{KL}(\rho_t \| \rho_0)} \leq & \gamma Z_5 \int_0^t \sqrt{\mathcal{L}_n(\rho_s, W_s)} \, \mathrm{d}s \\
\leq & \beta^{-1}(Z_5 A_2)t + \gamma^{-1} Z_5 A_1^{-1},
\end{aligned}
$$

thus concluding the proof.

## C.3. Proof of Corollary 4.9

*Proof of Corollary 4.9.* By Theorem 4.8, we have that, for $t > t_0$,

$$\mathcal{L}_{n,\lambda}(\rho_t, W_t) \leq \beta^{-1} C_4.$$

Then, by Lemma 4.3, we have that, for any $0 < \epsilon_0 < 1/2$, by picking

$$\beta \geq \max \left\{ (2C_1^2 n C_4)^{\frac{2}{\epsilon_0}}, \left( \frac{4q}{n} \right)^{\frac{1}{\epsilon_0}}, 64(qC_4)^2 \right\},$$

we have

$$\sigma_{\min}(W) \geq \beta^{-\epsilon_0}, \quad \sigma_{\max}(W)^2 \leq 2C_4.$$

Plugging this in (10) gives

$$\|\boldsymbol{E}_2(\epsilon_S, \beta; \gamma, \rho_t, W_t)\|_F^2 \leq 2nC_3^{-2}C_4\beta^{-1+2\epsilon_0} + (8\beta^4\gamma^4 C_1^2 C_4^2 + 2\beta^2 + 4\beta^{2+2\epsilon_0} C_4)C_1^2 n(\epsilon_S^t)^2.$$

By Lemma 4.6, we know that, if we pick $\epsilon_S$ small enough, there exists $T(\epsilon_S)$ s.t. for all $t > T(\epsilon_S)$ except a finite Lebesgue measure set,

$$\|\boldsymbol{E}_2(\epsilon_S, \beta; \gamma, \rho_t, W_t)\|_F^2 \leq 4nC_3^{-2}C_4\beta^{-1+2\epsilon_0}.$$

Consequently, taking

$$\beta \geq \left( 640 C_3^{-2} C_4^2 \frac{1}{\delta_0} \right)^{\frac{1}{1-2\epsilon_0}}$$

ensures that (16) is satisfied and, hence, we can apply (17) which gives that, for all $t > T(\epsilon_S)$ except a finite Lebesgue measure set,

$$NC1(H_{\rho_t}) \leq \frac{64nC_3^{-2}C_4\beta^{-1+2\epsilon_0}}{\frac{(q-1)n}{4qC_4} - 16nC_3^{-2}C_4\beta^{-1+2\epsilon_0}} \leq \delta_0.$$

Finally, by taking $\epsilon_0 = \frac{1}{3}$, we finish the proof.

$\square$

# D. Proofs in Section 5

Throughout this appendix, given $v, u \in \mathbb{R}^p$, we define the partial order $v \preceq u$ if $v_i \leq u_i$ for all $i \in [p]$. Additionally, if $u = R\mathbf{1}$, we write $v \preceq R$ as a shorthand. Similarly, we write $v \not\preceq u$ (resp. $v \not\preceq R$) if there exists some $i$ such that $v_i > u_i$ (resp. $v_i > R$). The symbols $\prec$ and $\not\prec$ are defined analogously. Furthermore, for a given vector $v$, we define the Jacobian as $J_R(v) = \text{diag}(\tau_R'(v_1), \ldots, \tau_R'(v_p)) \in \mathbb{R}^{p \times p}$.

## D.1. Proof of Theorem 5.3

We start with a result (proved in Appendix D.2) controlling the generalization error for data distributions that satisfy Assumptions 1 and 3. We recall that given a function class $\mathcal{F}$, the Rademacher complexity is defined as

$$\mathfrak{R}_n(\mathcal{F}) = \mathbb{E}_{\epsilon_i} \left[ \sup_{f \in \tilde{\mathcal{F}}} \frac{1}{n} \sum_{i=1}^n \epsilon_i w^\top \mathbb{E}_\rho[a\sigma(u^\top x_i)] \right],$$

where $\epsilon_i$ are i.i.d. Rademacher random variables.

**Lemma D.1.** *For $i \in \{1, \ldots, q\}$, let $\mathcal{F}_i$ be a class of functions from $\mathbb{R}^d \to \mathbb{R}$. Let $\mathcal{D}$ be a data distribution satisfies Assumptions 1 and 3, and let $x_1, \ldots, x_n \overset{i.i.d}{\sim} \mathcal{D}$. Then, for any $f : \mathbb{R}^d \to \mathbb{R}^q$ s.t. $[f]_i \in \mathcal{F}_i$, we have*

$$\boldsymbol{err}_{test}(f; \mathcal{D}) \leq \frac{2\sqrt{2}}{\sqrt{q}} \sqrt{\mathcal{L}_n(f)} + 4 \sum_{i=1}^q \mathfrak{R}_n(\mathcal{F}_i) + 6q\sqrt{\frac{\log(2/\delta)}{n}},$$

*with probability $> 1 - \delta$. Here, we define $\mathcal{L}_n(f) = \frac{1}{2n} \sum_{i=1}^n \|f(x_i) - y_i\|_2^2$.*

We then define the following functional class corresponding to the output function of our model:

$$\tilde{\mathcal{F}}(M_w, M_\rho) = \{f : \mathbb{R}^d \to \mathbb{R} \mid f(x) = w^\top h_\rho(x),\ \|w\|_2^2 \le M_w,\ \mathbb{E}_\rho[\|\theta\|_2^2] \le M_\rho\}. \tag{42}$$

The next lemma (proved in Appendix D.3) upper bounds the Rademacher complexity of the functional class in (42).

**Lemma D.2.** *Given $M_w, M_\rho > 0$, we have:*

$$\mathfrak{R}_n(\tilde{\mathcal{F}}(M_w, M_\rho)) \le \sqrt{\frac{M_w M_\rho C_1^2 \pi}{2n}}.$$

The next theorem (proved in Appendix D.4) shows that, if the stationary point we achieve has small regularized loss, then the test error is small.

**Theorem D.3.** *Let $(\rho, W)$ satisfy*

$$\mathcal{L}_{\lambda,n}(\rho, W) \le B\beta^{-1}(\log \beta)^\alpha.$$

*Then, for any $0 < \epsilon_0 < 1/2$ and*

$$\beta \ge \max\left\{ e^{\frac{4\alpha}{\epsilon_0} \log \frac{2\alpha}{\epsilon_0}}, (2C_1^2 nB)^{\frac{2}{\epsilon_0}}, \left(\frac{4q}{n}\right)^{\frac{1}{\epsilon_0}}, 64(qB)^2 \right\},$$

*the following upper bound on the test error holds*

$$\boldsymbol{err}_{test}(f(x; \rho, W); \mathcal{D}) \le 2\sqrt{2q^{-1}B}\beta^{-1/2}(\log \beta)^{\alpha/2} + 8qB(\log \beta)^\alpha \sqrt{\frac{C_1^2 \pi}{2n}} + 6q\sqrt{\frac{\log(2/\delta)}{n}},$$

*with probability $\ge 1 - \delta$.*

Theorem D.3 implies that it is necessary to control how $B$ scales with $n$ in order to control the generalization error. We now show that one can do so for $(\tau, M)$-linearly separable data.

**Lemma D.4.** *Let $\mathcal{D}$ be a bounded and $(\tau, M)$-linearly separable data distribution as per Definition 5.2. Pick $R > q$ and let $\sigma(z) = \frac{1}{1+e^{-z}}$. Then, for any $\epsilon > 0$, there exists $\widetilde{\rho}_1$ with*

$$\mathbb{E}_{\widetilde{\rho}_1}[\|\theta\|_2^2] \le q^2 + \frac{M^2(\log(\sqrt{q}/\epsilon))^2}{\tau^2} + \epsilon^2, \qquad \mathbb{E}_{\widetilde{\rho}_1}[\log \widetilde{\rho}_1] \le \frac{p+d}{2} \log\left(\frac{\epsilon^{-2}}{2\pi e}\right),$$

*such that $f_R(x; \widetilde{\rho}_1, W_0) = W_0^\top h_{\widetilde{\rho}_1}^R(x)$ approximates well the true data distribution:*

$$\mathbb{E}_{(x,y)\sim\mathcal{D}}[|f_R(x; \widetilde{\rho}_1, W_0) - y|^2] \le 2\epsilon^2 + 4(qd + C_0^2 p)C_1^2 \epsilon^2.$$

**Lemma D.5.** *Consider a bounded and $(\tau, M)$-linearly separable data distribution as per Definition 5.2. Let $(\rho_t, W_t)$ be obtained by **Stage 2** of Algorithm 1. Then, for any $\beta > 0$, we can pick $R$ large enough such that*

$$\mathcal{E}_n(\rho_t, W_t) \le C_8 \beta^{-1} \log \beta,$$

*with*

$$C_8 = 3\left(1 + 2(q^2 d + C_0^2 p)C_1 + \frac{M^2}{2\tau^2} + \frac{p+d}{4}\right) + \frac{p+d}{2}\log(2\pi).$$

The proofs of Lemma D.4 and Lemma D.5 are provided in Appendix D.5 and D.6 respectively. Lemma D.5 shows that the constant $B$ in the upper bound of the free energy does not blow up with $n$.

We are now ready to state and prove the full version of Theorem 5.3.

**Theorem D.6** (Full statement of Theorem 5.3). *Under Assumptions 1 and 3, let the data distribution be bounded and* $(\tau, M)$*-linearly separable as per Definition 5.2. Pick* $R > 1$ *large enough,* $n$ *large enough, and* $\beta = \left(640 C_1^2 n C_9^2 \frac{1}{\delta_0}\right)^6$. *Then, for any* $(\rho_t, W_t)$ *obtained by* **Stage 2** *of Algorithm 1, we have*

$$\boldsymbol{err}_{test}(f(\cdot; \rho_t, W_t); \mathcal{D}) \leq C_{10} \log(C_{11} n / \delta_0) \sqrt{\frac{1}{2n}} + 6q \sqrt{\frac{\log(2/\delta)}{n}}, \tag{43}$$

*with probability at least* $1 - \delta$. *Furthermore, there exist* $T(\beta)$ *s.t. for all* $t > T(\beta)$ *except a finite Lebesgue measure set,*

$$NC1(H_{\rho_t}) \leq \delta_0. \tag{44}$$

*The constants* $C_9, C_{10}, C_{11}$ *are given by*

$$C_9 = 9\left(2 + 4(d + C_0^2 p)C_1 + \frac{M^2}{2\tau^2} + \frac{p+d}{4}\right) + \frac{p+d}{2} + \frac{3(p+d)\log(2\pi)}{2} + 2(1 + (p+d)\log 8\pi),$$

$$C_{10} = 50 q C_9 \sqrt{C_1^2 \pi},$$

$$C_{11} = 640 C_1^2 C_9^2.$$

*Proof of Theorem 5.3.* By Lemma A.3 and Lemma D.5, we have

$$\mathcal{L}_{\lambda,n}(\rho_t, W_t) \leq 3\mathcal{E}_n(\rho_t, W_t) + \beta^{-1}\frac{p+d}{2}\log\beta + 2\beta^{-1}(1 + (p+d)\log 8\pi) \leq C_9 \beta^{-1}\log\beta,$$

with

$$C_9 = 3C_8 + \frac{p+d}{2} + 2(1 + (p+d)\log 8\pi).$$

Thus by Theorem D.3,

$$\boldsymbol{err}_{test}(f(x; \rho_t, W_t); \mathcal{D}) \leq 16(C_9 \beta^{-1}\log\beta)^{\frac{1}{2}} + 8q C_9 \log\beta \sqrt{\frac{C_1^2 \pi}{2n}} + 6q \sqrt{\frac{\log(2/\delta)}{n}}, \tag{45}$$

with probability $\geq 1 - \delta$. By using that $\beta = \left(640 C_1^2 n C_9^2 \frac{1}{\delta_0}\right)^6$ and that $n$ is large enough, the desired bound (43) readily follows. Finally, by proceeding as in the argument of Corollary 4.9, we also obtain (44) and the proof is complete. $\square$

### D.2. Proof of Lemma D.1

*Proof of Lemma D.1.* It is easy to see that $\boldsymbol{err}_{test}(f; \mathcal{D}) = \frac{1}{q}\sum_{k=1}^{q} \mathbb{E}_{\mathcal{D}(\cdot | e_k)}[\mathbf{1}_{\text{One-Hot}(f(x)) \neq e_k}]$. We have the following upper bound:

$$\begin{aligned}
\mathbf{1}_{\text{One-Hot}(f(x)) \neq e_k} &= \mathbf{1}_{\text{One-Hot}(2f(x)-1) \neq \text{One-Hot}(2e_k - 1)} \\
&\leq \mathbf{1}_{(2f(x)-1) \odot (2e_k - 1) \not\geq 0} \quad \text{(Note that } 2e_k[i] - 1 \in \{\pm 1\}) \\
&\leq \sum_{i=1}^{q} \mathbf{1}_{(2[f(x)]_i - 1)(2e_k[i] - 1) < 0},
\end{aligned}$$

where by $\mathbf{1}_E$ here denote the indicator function of an event $E$.

We now follow the approach of (Chen et al., 2020, Lemma 5.6) and define the surrogate loss

$$\ell_{ramp}(y', y) = \begin{cases} 1, & yy' < 0; \\ -2yy' + 1, & 0 \leq yy' < 1/2; \\ 0, & yy' \geq 1/2. \end{cases}$$

Then, $\ell_{ramp}$ is 2-Lipschitz in $y'$ and, for $y \in \{\pm 1\}$,

$$\mathbf{1}_{yy' < 0} \leq \ell_{ramp}(y', y) \leq |y - y'|.$$

Thus, we have that, for any $x$,

$$\sum_{i=1}^{q} \mathbf{1}_{(2[f(x)]_i - 1)(2e_k[i] - 1) < 0} \leq \sum_{i=1}^{q} \ell_{ramp}(2[f(x)]_i - 1, 2e_k[i] - 1) \leq 2 \sum_{i=1}^{q} |[f(x)]_i - e_k[i]| \leq 2\sqrt{q\|f(x) - e_k\|_2^2},$$

which leads to the following generalization bound

$$
\begin{aligned}
\mathbf{err}_{test}(f; \mathcal{D}) &= \frac{1}{q} \sum_{k=1}^{q} \mathbb{E}_{\mathcal{D}(\cdot|e_k)}[\mathbf{1}_{\text{One-Hot}(f(x)) \neq e_k}] \\
&\leq \frac{1}{q} \sum_{k=1}^{q} \mathbb{E}_{\mathcal{D}(\cdot|e_k)}\left[\sum_{i=1}^{q} \mathbf{1}_{(2[f(x)]_i - 1)(2e_k[i] - 1) < 0}\right] \\
&\leq \frac{2}{q} \sum_{k=1}^{q} \mathbb{E}_{\mathcal{D}(\cdot|e_k)}\left[\sum_{i=1}^{q} \ell_{ramp}(2[f(x)]_i - 1, 2e_k[i] - 1)/2\right] \\
&\leq \frac{2}{qn} \sum_{j=1}^{n} \left[\sum_{i=1}^{q} \ell_{ramp}(2[f(x_j)]_i - 1, 2y_j[i] - 1)/2\right] + 4\sum_{i=1}^{q} \mathfrak{R}_n(\mathcal{F}_i) + 6q\sqrt{\frac{\log(2/\delta)}{n}} \quad \text{with probability} > 1 - \delta \\
&\leq \sum_{j=1}^{n} \frac{2}{n\sqrt{q}} \|f(x_j) - y_j\|_2 + 4\sum_{i=1}^{q} \mathfrak{R}_n(\mathcal{F}_i) + 6q\sqrt{\frac{\log(2/\delta)}{n}} \quad \text{with probability} > 1 - \delta \\
&\leq \frac{2\sqrt{2}}{\sqrt{q}} \sqrt{\frac{1}{2n} \sum_{j=1}^{n} \|f(x_j) - y_j\|_2^2} + 4\sum_{i=1}^{q} \mathfrak{R}_n(\mathcal{F}_i) + 6q\sqrt{\frac{\log(2/\delta)}{n}} \quad \text{with probability} > 1 - \delta \\
&\leq \frac{2\sqrt{2}}{\sqrt{q}} \sqrt{\mathcal{L}_n(f)} + 4\sum_{i=1}^{q} \mathfrak{R}_n(\mathcal{F}_i) + 6q\sqrt{\frac{\log(2/\delta)}{n}} \quad \text{with probability} > 1 - \delta.
\end{aligned}
$$

$\square$

### D.3. Proof of Lemma D.2

*Proof of Lemma D.2.* The proof is a modification from (Takakura & Suzuki, 2024, Lemma 4.3). We first use the fact that the Rademacher complexity can be upper bounded by Gaussian complexity, which is defined as

$$\mathfrak{G}_n(\tilde{\mathcal{F}}(M_w, M_\rho)) = \mathbb{E}_{\epsilon_i}\left[\sup_{f \in \tilde{\mathcal{F}}(M_w, M_\rho)} \frac{1}{n} \sum_{i=1}^{n} \epsilon_i w^\top \mathbb{E}_\rho[a\sigma(u^\top x_i)]\right],$$

with $\epsilon_i \overset{i.i.d}{\sim} \mathcal{N}(0, 1)$. For any function class $\mathcal{F}$ and any $n$, we have (Wainwright, 2019, section 5.2):

$$\mathfrak{R}_n(\mathcal{F}) \leq \sqrt{\frac{\pi}{2}} \mathfrak{G}_n(\mathcal{F}).$$

Thus it is sufficient to upper bound the Gaussian complexity of the function class:

$$\mathfrak{G}_n(\tilde{\mathcal{F}}(M_w, M_\rho)) = \mathbb{E}_{\epsilon_i}\left[\sup_{f \in \tilde{\mathcal{F}}(M_w, M_\rho)} \frac{1}{n}\sum_{i=1}^{n} \epsilon_i w^\top \mathbb{E}_\rho[a\sigma(u^\top x_i)]\right]$$

$$= \mathbb{E}_{\epsilon_i}\left[\sup_{(\rho,w):\|w\|_2^2 \leq M_w, \mathbb{E}_\rho[\|\theta\|_2^2] \leq M_\rho} \frac{1}{n}\sum_{i=1}^{n} \epsilon_i w^\top \mathbb{E}_\rho[a\sigma(u^\top x_i)]\right]$$

$$\leq \mathbb{E}_{\epsilon_i}\left[\sup_{(\rho,w):\|w\|_2^2 \leq M_w, \mathbb{E}_\rho[\|\theta\|_2^2] \leq M_\rho} \|w\|_2 \left\|\frac{1}{n}\sum_{i=1}^{n} \epsilon_i \mathbb{E}_\rho[a\sigma(u^\top x_i)]\right\|_2\right]$$

$$\leq \sqrt{M_w}\mathbb{E}_{\epsilon_i}\left[\sup_{\rho:\mathbb{E}_\rho[\|\theta\|_2^2] \leq M_\rho} \left\|\frac{1}{n}\sum_{i=1}^{n} \epsilon_i \mathbb{E}_\rho[a\sigma(u^\top x_i)]\right\|_2\right]$$

$$= \sqrt{\frac{M_w}{n}}\mathbb{E}_{\epsilon_i}\left[\sup_{\rho:\mathbb{E}_\rho[\|\theta\|_2^2] \leq M_\rho} \|\mathbb{E}_\rho[aZ(u)]\|_2\right], \quad \text{where we define } Z(u) = \frac{1}{\sqrt{n}}\sum_{i=1}^{n} \epsilon_i \sigma(u^\top x_i)$$

$$\leq \sqrt{\frac{M_w}{n}}\mathbb{E}_{\epsilon_i}\left[\sup_{\rho:\mathbb{E}_\rho[\|\theta\|_2^2] \leq M_\rho} \sqrt{\mathbb{E}_\rho[a^\top a' Z(u)Z(u')]}\right]$$

$$\leq \sqrt{\frac{M_w}{n}}\mathbb{E}_{\epsilon_i}\left[\sup_{\rho:\mathbb{E}_\rho[\|\theta\|_2^2] \leq M_\rho} \left(\mathbb{E}_\rho[(a^\top a')^2]\mathbb{E}_\rho[Z(u)^2 Z(u')^2]\right)^{1/4}\right]$$

$$\leq \sqrt{\frac{M_w}{n}}\mathbb{E}_{\epsilon_i}\left[\sup_{\rho:\mathbb{E}_\rho[\|\theta\|_2^2] \leq M_\rho} \left(\mathbb{E}_\rho[\|a\|_2^2]\mathbb{E}_\rho[\|a'\|_2^2]\mathbb{E}_\rho[Z(u)^2]\mathbb{E}_\rho[Z(u')^2]\right)^{1/4}\right]$$

$$= \sqrt{\frac{M_w}{n}}\mathbb{E}_{\epsilon_i}\left[\sup_{\rho:\mathbb{E}_\rho[\|\theta\|_2^2] \leq M_\rho} \left(\mathbb{E}_\rho[\|a\|_2^2]\mathbb{E}_\rho[Z(u)^2]\right)^{1/2}\right]$$

$$\leq \sqrt{\frac{M_w M_\rho}{n}}\mathbb{E}_{\epsilon_i}\left[\sup_{\rho:\mathbb{E}_\rho[\|\theta\|_2^2] \leq M_\rho} \sqrt{\mathbb{E}_\rho[Z(u)^2]}\right].$$

Note that $Z(u) \sim \mathcal{N}(0, \phi(u)^2)$ with $\phi(u) = \frac{1}{n}\sum_{i=1}^{n} \sigma(u^\top x_i)^2 \leq C_1^2$. Thus,

$$\mathfrak{G}_n(\tilde{\mathcal{F}}(M_w, M_\rho)) \leq \sqrt{\frac{M_w M_\rho C_1^2}{n}},$$

and the desired claim readily follows. □

## D.4. Proof of Theorem D.3

*Proof of Theorem D.3.* By Lemma 4.3, we have that

$$\sigma_{\min}(W) \geq \beta^{-\epsilon_0}, \qquad \|W\|_F^2 \leq 2B(\log\beta)^\alpha.$$

Recall that $\mathcal{L}_{\lambda,n}(\rho, W) \leq B\beta^{-1}(\log\beta)^\alpha$. Thus,

$$\mathbb{E}_\rho[\|\theta\|_2^2] \leq 2\beta\mathcal{L}_{\lambda,n}(\rho, W) \leq 2B(\log\beta)^\alpha.$$

Consequently, for each $i$, $[f(x; \rho, W)]_i \in \tilde{F}(M_w M_\rho)$ with

$$M_w = 2B(\log\beta)^\alpha, \qquad M_\rho = 2B(\log\beta)^\alpha.$$

Hence, combining Lemma D.1 and Lemma D.2, we get

$$\mathbf{err}_{test}(f(x, \rho, W); \mathcal{D}) \leq 2\sqrt{2q^{-1}B}\beta^{-1/2}(\log\beta)^{\alpha/2} + 8qB(\log\beta)^\alpha\sqrt{\frac{C_1^2\pi}{2n}} + 6q\sqrt{\frac{\log(2/\delta)}{n}}.$$

with probability $> 1 - \delta$, which completes the proof. □

## D.5. Proof of Lemma D.4

*Proof of Lemma D.4.* Using the definition of linearly separable data, we first show that, for each $k$, we could use one neuron to approximate the perfect classifier. In particular, having fixed $k$, let $a_k = qe_k$, $u_k = \frac{\log(\sqrt{q}/\epsilon)}{\tau}\hat{u}_k$. Then, we have

$$\sigma(u_k^\top x) \geq \frac{1}{1 + \exp(-\log(\sqrt{q}/\epsilon))} = \frac{1}{1 + \epsilon/\sqrt{q}}, \quad \text{for } x \in \text{supp}(\mathcal{D}(\cdot|e_k)),$$

$$\sigma(u_k^\top x) < \frac{1}{1 + \exp(\log(\sqrt{q}/\epsilon))} = \frac{\epsilon/\sqrt{q}}{1 + \epsilon/\sqrt{q}}, \quad \text{for } x \in \text{supp}(\mathcal{D}(\cdot|e_{k'})) \text{ with } k' \neq k.$$

Define $\rho_\circ = \left(\frac{1}{q}\sum_{k=1}^q \delta_{(a_k, u_k)}\right)$ and $\hat{\rho}_\circ = \rho_\circ * \gamma_\zeta$, where $\gamma_\zeta \sim \exp(-\|\cdot\|_2^2/(2\zeta^2))$. By picking $R > q$, we know that, under the distribution $\rho_\circ$, $\tau_R(a) = a$ and thus:

$$h_{\rho_\circ}^R(x) = h_{\rho_\circ}(x) = p_{k,k}e_k + \sum_{k' \neq k} p_{k',k}e_{k'},$$

for any $x \in \text{supp}(\mathcal{D}(\cdot|e_k))$ with $p_{k,k} \geq \frac{1}{1+\epsilon/\sqrt{q}}, p_{k',k} \leq \frac{\epsilon/\sqrt{q}}{1+\epsilon/\sqrt{q}}$. This implies that:

$$\mathbb{E}_\mathcal{D}[\|f_R(x; \rho_\circ, W_0) - y\|_2^2] = \|p_{k,k}e_k + \sum_{k' \neq k} p_{k',k}e_{k'} - e_k\|_2^2 \leq \epsilon^2.$$

We then have the following upper bounds:

$$\|h_{\hat{\rho}_\circ(x)}^R - h_{\rho_\circ}(x)\|_2^2$$

$$= \left\|\mathbb{E}_{\rho_\circ}\mathbb{E}_G[\tau_R(a + \zeta G_a)\sigma((u + \zeta G_u)^\top x)] - \mathbb{E}_{\rho_\circ}[a\sigma(u^\top x)]\right\|_2^2 \quad \text{where } (G_a, G_u) \sim \mathcal{N}(0, I_{p+d+1})$$

$$= \left\|\mathbb{E}_{\rho_\circ}\mathbb{E}_G[(\tau_R(a) + \zeta J_R(\tilde{a})G_a)\sigma((u + \zeta G_u)^\top x)] - \mathbb{E}_{\rho_\circ}[a\sigma(u^\top x)]\right\|_2^2$$

$$\leq 2\left\|\mathbb{E}_{\rho_\circ}\mathbb{E}_G[\tau_R(a)\sigma((u + \zeta G_u)^\top x)] - \mathbb{E}_{\rho_\circ}[a\sigma(u^\top x)]\right\|_2^2 + 2\left\|\mathbb{E}_{\rho_\circ}\mathbb{E}_G[\zeta J_R(\tilde{a})G_a\sigma((u + \zeta G_u)^\top x)]\right\|_2^2.$$

For the first term, we have

$$\left\|\mathbb{E}_{\rho_\circ}\mathbb{E}_G[\tau_R(a)\sigma((u + \zeta G_u)^\top x)] - \mathbb{E}_{\rho_\circ}[a\sigma(u^\top x)]\right\|_2^2 = \left\|\mathbb{E}_{\rho_\circ}\mathbb{E}_G[a\sigma((u + \zeta G_u)^\top x)] - \mathbb{E}_{\rho_\circ}[a\sigma(u^\top x)]\right\|_2^2$$

$$= \sum_{k=1}^q |\mathbb{E}_G\sigma((u_k + \zeta G_u)^\top x) - \sigma(u_k^\top x)|^2$$

$$\leq qC_1^2\left(\mathbb{E}_G|\zeta G_u^\top x|\right)^2 \leq qC_1^2 d\zeta^2.$$

For the second term, we have

$$\left\|\mathbb{E}_{\rho_\circ}\mathbb{E}_G[\zeta J_R(\tilde{a})G_a\sigma((u + \zeta G_u)^\top x)]\right\|_2^2 \leq \zeta^2 C_0^2 C_1^2 p.$$

Thus, we have the following bounds on the test error:

$$\mathbb{E}_\mathcal{D}\left[|f_R(x; \hat{\rho}_\circ, W_0) - y|^2\right] \leq 2\mathbb{E}_\mathcal{D}\left[|f_R(x; \rho_\circ, W_0) - y|^2\right] + 2\mathbb{E}_\mathcal{D}\left[|f_R(x; \rho_\circ, W_0) - f_R(x; \hat{\rho}_\circ, W_0)|^2\right]$$

$$\leq 2\epsilon^2 + 4(qd + C_0^2 p)C_1^2\zeta^2.$$

Finally, we need to upper bound the second moment and the negative entropy of $(\hat{\rho}_\circ, W_0)$. For the second moment term, a direct computation gives:

$$\mathbb{E}_{\hat{\rho}_\circ}[\|\theta\|_2^2] \leq q^2 + \frac{M^2(\log(\sqrt{q}/\epsilon))^2}{\tau^2} + \zeta^2.$$

For the entropy term, we have:

$$\mathbb{E}_{\hat{\rho}_\circ}\log(\hat{\rho}_\circ) \leq \mathbb{E}_{\gamma_\zeta}\log(\gamma_\zeta) = \frac{p+d}{2}\log\left(\frac{\zeta^{-2}}{2\pi e}\right).$$

Taking $\zeta = \epsilon$ and $\tilde{\rho}_1 = \hat{\rho}_\circ$ concludes the proof. $\qquad\square$

## D.6. Proof of Lemma D.5

We first upper bound the approximated free energy at $(\rho_1, W_0)$ obtained by **Stage 1** of Algorithm 1 using the optimality of $\rho_1, W_0$.

$$\mathcal{E}_n^R(\rho_1, W_0) \leq \mathcal{L}_n^R(\widetilde{\rho}_1, W_0) + \frac{\beta^{-1}}{2}\|W_0\|_F^2 + \frac{\beta^{-1}}{2}\mathbb{E}_{\widetilde{\rho}_1}[\|a\|_2^2 + \|u\|_2^2] + \beta^{-1}\mathbb{E}_{\widetilde{\rho}_1}\log\widetilde{\rho}_1.$$

Next, by the construction in Lemma D.4 with $\epsilon^2 = \beta^{-1}$ and by letting $\mathcal{D}$ be the empirical distribution of training samples, we have

$$\mathcal{L}_n^R(\widetilde{\rho}_1, W_0) \leq (1 + 2(q^2 d + C_0^2 p)C_1)\beta^{-1}$$

$$\mathbb{E}_{\widetilde{\rho}_1}[\|\theta\|_2^2] \leq q^2 + \frac{M^2 \log(q\beta)^2}{4\tau^2} + \beta^{-1}$$

$$\mathbb{E}_{\widetilde{\rho}_1}[\log\widetilde{\rho}_1] = \frac{p+d}{2}\log(\beta/2\pi e),$$

which implies that

$$\mathcal{E}_n^R(\rho_1, W_0) \leq C_6\beta^{-1}\log\beta,$$
$$\mathcal{L}_n^R(\rho_1, W_0) + \frac{\beta^{-1}}{2}\|W_0\|_F^2 \leq C_7\beta^{-1}\log\beta, \tag{46}$$

with $C_6 = \left(1 + 2(q^2 d + C_0^2 p)C_1 + \frac{M^2}{2\tau^2} + \frac{p+d}{4}\right)$ and $C_7 = C_6 + \frac{p+d}{2} + \log 2\pi$.

To conclude, we need an upper bound on $\mathcal{E}_n(\rho_1, W_0)$, which is obtained using the following intermediate result (proved in Appendix D.7) that controls the approximation error caused by the approximated second-layer.

**Lemma D.7.** *Let $(\rho_1, W_0)$ be the unique global minimizer achieved by* Phase 1 *of Algorithm 1. Then, it holds that*

$$\|H_{\rho_1} - H_{\rho_1}^R\|_F^2 \leq (4C_6 \log\beta + 4(1 + (p+d)\log 8\pi))\frac{C_1^2 p^2 \sqrt{2}}{\sqrt{\pi}R}$$

$$\cdot \exp\left(2\sqrt{2pC_7\beta\log\beta}C_1(R + C_0)\sigma_{\max}(W_0) - \frac{R^2}{2}\right).$$

*In particular, for any fixed $\beta$, we have*

$$\lim_{R\to+\infty}\|H_{\rho_1} - H_{\rho_1}^R\|_F = 0.$$

Finally, to bound $\mathcal{E}_n(\rho_1, W_0)$, we write

$$\mathcal{E}_n(\rho_1, W_0) = \mathcal{E}_n^R(\rho_1, W_0) + \mathcal{L}_n(\rho_1, W_0) - \mathcal{L}_n^R(\rho_1, W_0)$$

$$\leq \mathcal{E}_n^R(\rho_1, W_0) + \frac{1}{2n}\left(\|W_0^\top H_{\rho_1} - Y\|_F^2 - \|W_0^\top H_{\rho_1}^R - Y\|_F^2\right)$$

$$\leq \mathcal{E}_n^R(\rho_1, W_0) + \frac{1}{n}\|W_0^\top H_{\rho_1} - W_0^\top H_{\rho_1}^R\|_F^2 + \frac{1}{2n}\|W_0^\top H_{\rho_1}^R - Y\|_F^2$$

$$\leq \mathcal{E}_n^R(\rho_1, W_0) + \frac{\sigma_{\max}(W_0)^2}{n}\|H_{\rho_1} - H_{\rho_1}^R\|_F^2 + \mathcal{L}_n^R(\rho_1, W_0)$$

$$\leq 2\mathcal{E}_n^R(\rho_1, W_0) + \beta^{-1}\frac{p+d}{2}\log(2\pi) + \frac{1}{n}\|H_{\rho_1} - H_{\rho_1}^R\|_F^2,$$

where the last line follows from Lemma A.3. By Lemma D.7 we have that $\lim_{R\to\infty}\|H_\rho - H_\rho^R\|_F^2 = 0$, which means that one can pick $R$ large enough such that $\frac{1}{n}\|H_\rho - H_\rho^R\|_F^2 \leq \mathcal{E}_n^R(\rho_1, W_0)$. Noting that the free energy is non-increasing when running **Stage 2** of Algorithm 1 concludes the proof.

## D.7. Proof of Lemma D.7

*Proof of Lemma D.7.* In the proof, we will write $(\rho, W)$ instead of $(\rho_1, W_0)$ given that there is no confusion. We start by showing that

$$\int_{|a|\not\leq R}\rho_1(a, u)\,\mathrm{d}a\mathrm{d}u \leq \frac{p\sqrt{2}}{\sqrt{\pi}R}\exp\left(2\sqrt{2pC_7\beta\log\beta}C_1(R + C_0) - \frac{R^2}{2}\right). \tag{47}$$

Recall from Proposition 5.1 that $\rho$ has the Gibbs form in (27), i.e.,

$$\rho(a,u) = Z_R(\rho)^{-1} \exp\left(-\frac{\beta}{n}\tau_R(a)^\top W(W^\top H_\rho^R - Y)\sigma(X^\top u) - \frac{1}{2}(\|a\|_2^2 + \|u\|_2^2)\right),$$

where $Z_R(\rho)$ denotes the normalization constant. Thus, we have

$$\int_{|a|\not\preceq R} \rho(a,u)\,\mathrm{d}a\mathrm{d}u$$

$$= Z_R(\rho)^{-1} \int_{|a|\not\preceq R} \exp\left(-\frac{\beta}{n}\tau_R(a)^\top W(W^\top H_\rho^R - Y)\sigma(X^\top u) - \frac{1}{2}(\|a\|_2^2 + \|u\|_2^2)\right)\,\mathrm{d}a\mathrm{d}u$$

$$\le Z_R(\rho)^{-1}\left\{\sup_{a,u}\exp\left(-\frac{\beta}{n}\tau_R(a)^\top W(W^\top H_\rho^R - Y)\sigma(X^\top u)\right)\right\}\int_{|a|\not\preceq R} \exp\left(-\frac{1}{2}(\|a\|_2^2 + \|u\|_2^2)\right)\,\mathrm{d}a\mathrm{d}u.$$

We upper bound the various terms separately. First, we have

$$\sup_{a,u}\left|\frac{\beta}{n}\tau_R(a)^\top W(W^\top H_\rho^R - Y)\sigma(X^\top u)\right| \le \frac{\beta}{n}\sigma_{\max}(W)\|\tau_R(a)\|_2\|W^\top H_\rho^R - Y\|_F\|\sigma(X^\top u)\|_2$$

$$\le \frac{\beta}{n}(R+C_0)C_1\sqrt{np}\sigma_{\max}(W)\|W^\top H_\rho^R - Y\|_F$$

$$= \beta\sqrt{2p\mathcal{L}_n^R(\rho,W)}C_1(R+C_0)\sigma_{\max}(W)$$

$$\le \beta\sqrt{2pC_7\beta^{-1}\log\beta}C_1(R+C_0)\sigma_{\max}(W),$$

where in the last passage we use (46).

Next, we lower bound the normalization constant as

$$Z_R(\rho) = \int \exp\left(-\frac{\beta}{n}\tau_R(a)^\top W(W^\top H_\rho^R - Y)\sigma(X^\top u) - \frac{1}{2}(\|a\|_2^2 + \|u\|_2^2)\right)\,\mathrm{d}a\mathrm{d}u$$

$$\ge \exp\left(-\sup_{a,u}\left|\frac{\beta}{n}\tau_R(a)^\top W(W^\top H_\rho^R - Y)\sigma(X^\top u)\right|\right)\int \exp\left(-\frac{1}{2}(\|a\|_2^2 + \|u\|_2^2)\right)\mathrm{d}a\mathrm{d}u$$

$$\ge \exp\left(-\beta\sqrt{2pC_7\beta^{-1}\log\beta}C_1(R+C_0)\sigma_{\max}(W)\right)\int \exp\left(-\frac{1}{2}(\|a\|_2^2 + \|u\|_2^2)\right)\mathrm{d}a\mathrm{d}u\mathrm{d}b_0$$

$$= \exp\left(-\beta\sqrt{2pC_7\beta^{-1}\log\beta}C_1(R+C_0)\sigma_{\max}(W)\right)\left(\frac{1}{2\pi}\right)^{-(p+d)/2}.$$

Finally, we bound

$$\int_{|a|\not\preceq R} \exp\left(-\frac{1}{2}(\|a\|_2^2 + \|u\|_2^2)\right)\,\mathrm{d}a\mathrm{d}u$$

$$\le p\int_{|a_1|>R} \exp\left(-\frac{1}{2}|a_1|^2\right)\,\mathrm{d}a_1 \int \exp\left(-\frac{1}{2}(\sum_{i=2}^p |a_i|^2 + \|u\|_2^2)\right)\,\mathrm{d}a_2\ldots\mathrm{d}a_p\mathrm{d}u$$

$$\le 2p\left(\frac{1}{2\pi}\right)^{-(p+d)/2}\int_R^{+\infty} \exp\left(-\frac{1}{2}|a_1|^2\right)\,\mathrm{d}a_1$$

$$\le 2p\left(\frac{1}{2\pi}\right)^{-(p+d)/2}\frac{1}{R}\int_R^{+\infty} a_1\exp\left(-\frac{1}{2}|a_1|^2\right)\,\mathrm{d}a_1$$

$$= 2p\left(\frac{1}{2\pi}\right)^{-(p+d)/2}\frac{\exp\left(-\frac{R^2}{2}\right)}{R}.$$

Combining all the bounds , the desired result (47) follows.

The following chain of inequalities holds

$$
\begin{aligned}
\|H_\rho - H_\rho^R\|_F^2 &\leq \|\mathbb{E}_\rho[|a\sigma(u^\top X)|\mathbf{1}_{|a|\not\leq R}]\|_F^2 \\
&= \mathbb{E}_{\theta,\theta'\sim\rho}[|a|^\top|a'||\sigma(u^\top X)||\sigma(X^\top u')|\mathbf{1}_{|a|\not\leq R}\mathbf{1}_{|a'|\not\leq R}] \quad (\theta' \text{ is an independent copy of } \theta) \\
&\leq \sqrt{\mathbb{E}_{\theta,\theta'\sim\rho}[(|a|^\top|a'|)^2\mathbf{1}_{|a|\not\leq R}\mathbf{1}_{|a'|\not\leq R}]}\sqrt{\mathbb{E}_{\theta,\theta'\sim\rho}[(|\sigma(u^\top X)||\sigma(X^\top u')|)^2\mathbf{1}_{|a|\not\leq R}\mathbf{1}_{|a'|\not\leq R}]} \\
&\leq \mathbb{E}_\rho[\|a\|_2^2\mathbf{1}_{|a|\not\leq R}]\mathbb{E}[\|\sigma(X^\top u)\|_2^2\mathbf{1}_{|a|\not\leq R}] \\
&\leq \mathbb{E}_\rho[\|a\|_2^2\mathbf{1}_{|a|\not\leq R}]C_1^2 p \int_{|a|\not\leq R} \rho(a,u)\,\mathrm{d}a\mathrm{d}u \\
&\leq \mathbb{E}_\rho[\|a\|_2^2\mathbf{1}_{|a|\not\leq R}]\frac{C_1^2 p^2\sqrt{2}}{\sqrt{\pi}R}\exp\left(2\beta\sqrt{2C_7 p\beta^{-1}\log\beta}C_1(R+C_0)\sigma_{\max}(W) - \frac{R^2}{2}\right),
\end{aligned}
$$

where the last passage follows from (47). Finally, we upper bound $\mathbb{E}_\rho[\|a\|_2^2\mathbf{1}_{|a|\not\leq R}]$ as

$$
\begin{aligned}
\mathbb{E}_\rho[\|a\|_2^2\mathbf{1}_{|a|\not\leq R}] &\leq \mathbb{E}_\rho[\|a\|_2^2] \\
&\leq \mathbb{E}_\rho[\|a\|_2^2 + \|u\|_2^2] \\
&\leq 4\beta\mathcal{E}_n^R(\rho, W) + 4(1+(p+d)\log 8\pi) \\
&\leq 4\beta\left(C_6\beta^{-1}\log\beta\right) + 4(1+(p+d)\log 8\pi),
\end{aligned}
$$

where the third line follows from Lemma A.3 and the fourth line from (46). Combining these two bounds gives the desired result.

$\square$

# E. Additional Numerical Results

## E.1. Extra Experiments on Three-layer Networks

Figure 2 plots the normalized balancedness metric $NB(\rho, W)$ defined in (25) during training. The results clearly show that the network does not become balanced at convergence.

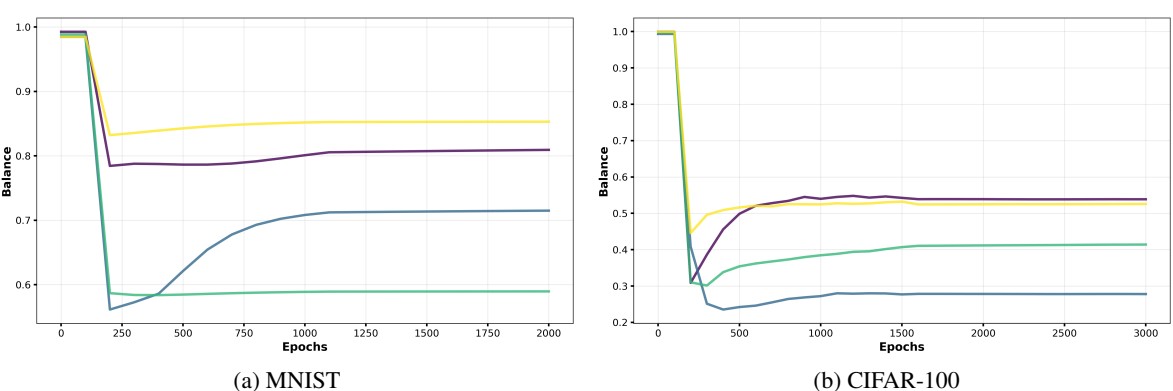

(a) MNIST        (b) CIFAR-100

Figure 2: Normalized balancedness (see (25)) as a function of the number of training epochs, with each color representing an independent experiment.

Figure 3 plots the approximately linear relation between the $\log$ of the NC1 metric and either the $\log$ of the gradient norm (Figure 3a) or of the training loss (Figure 3b). This clearly shows the polynomial relation between the gradient norm/training loss and NC1 as in Corollary 4.4.

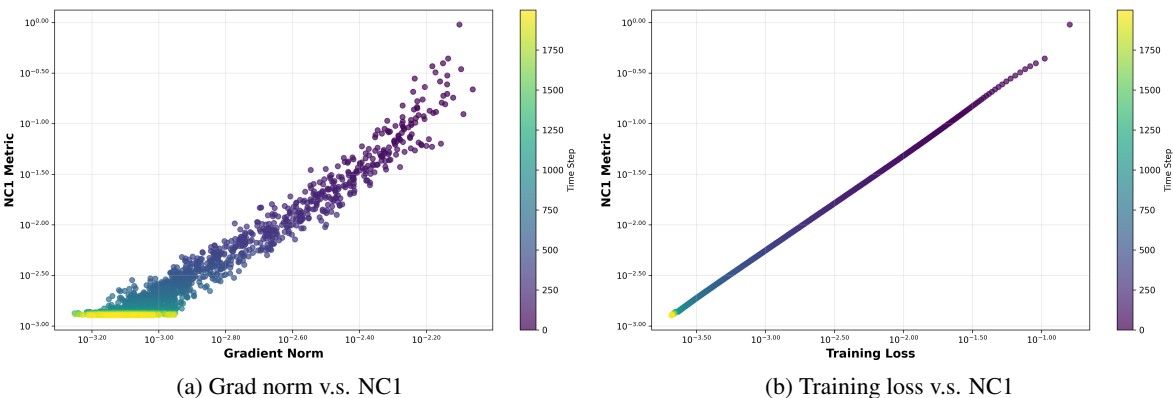

(a) Grad norm v.s. NC1                  (b) Training loss v.s. NC1

Figure 3: Scatterplot of average gradient norm/training loss and average NC1 for training a three-layer network on MNIST. The color bar indicates the training time step.

## E.2. Experiments on ResNet-18 and VGG-11

Figure 4 plots the evolution of training loss, gradient norm and NC1 metric during training for ResNet-18 (Figure 4a) and VGG-11 (Figure 4b). The curve indicates that the decrease of the NC1 metric with training loss and gradient norm (as predicted by Corollary 4.4) also holds for real networks.

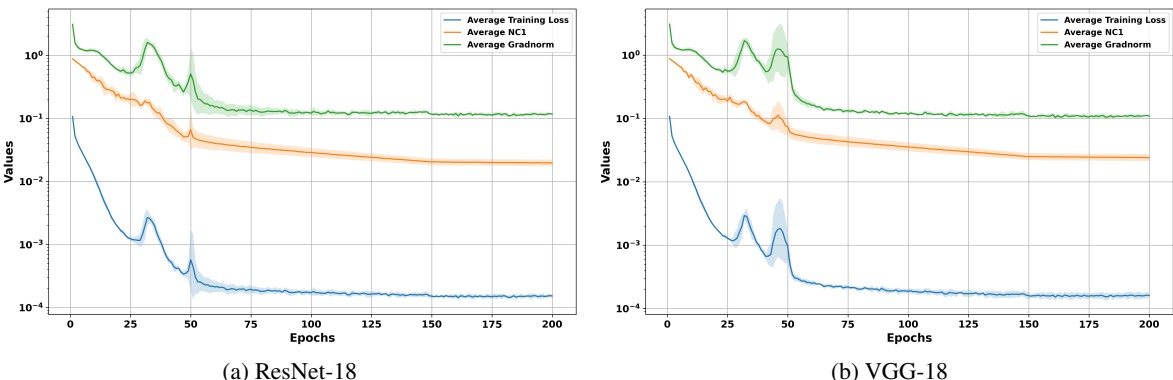

(a) ResNet-18                  (b) VGG-18

Figure 4: Average training loss (blue), NC1 (orange) and gradient norm (green) during SGD training for ResNet-18 and VGG-11 on CIFAR-10. We report the average for 4 independent experiments, as well as the confidence interval at 1 standard deviation.

Figures 5 and 6 plot the relation between the $\log$ of gradient norm (Figures 5a and 6a), training loss (Figures 5b and 6b) and $\log$ of NC1. In the terminal phase, the $\log$ of gradient norm and training loss have an approximately linear relation with the $\log$ of NC1, which implies a polynomial relation between the gradient norm/training loss and NC1 as in three-layer networks. However, the early phase of training exhibits a different non-linear relation compared to three-layer networks.

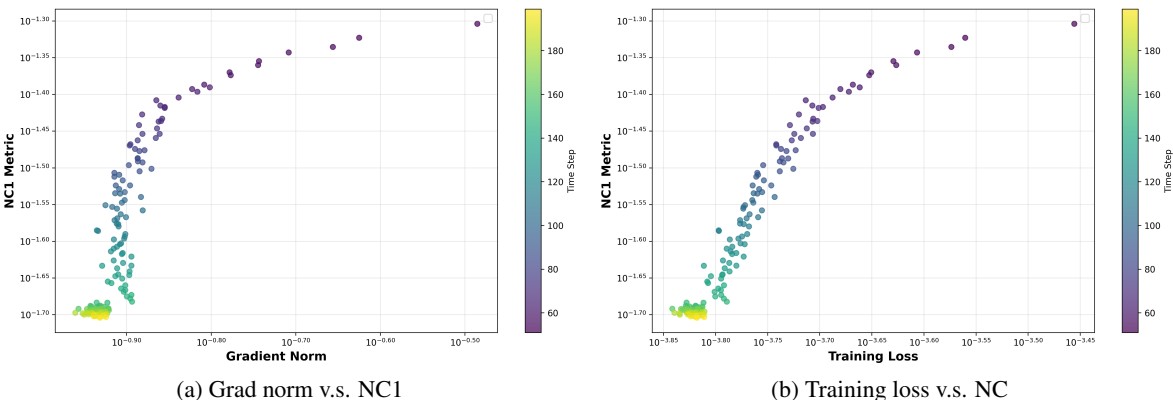

(a) Grad norm v.s. NC1

(b) Training loss v.s. NC

Figure 5: Scatterplot of gradient norm/training loss and NC1 for training ResNet-18 on CIFAR-10. The plot starts from epoch 51, right after the learning rate is reduced by a factor 10. The color bar indicates the training time step.

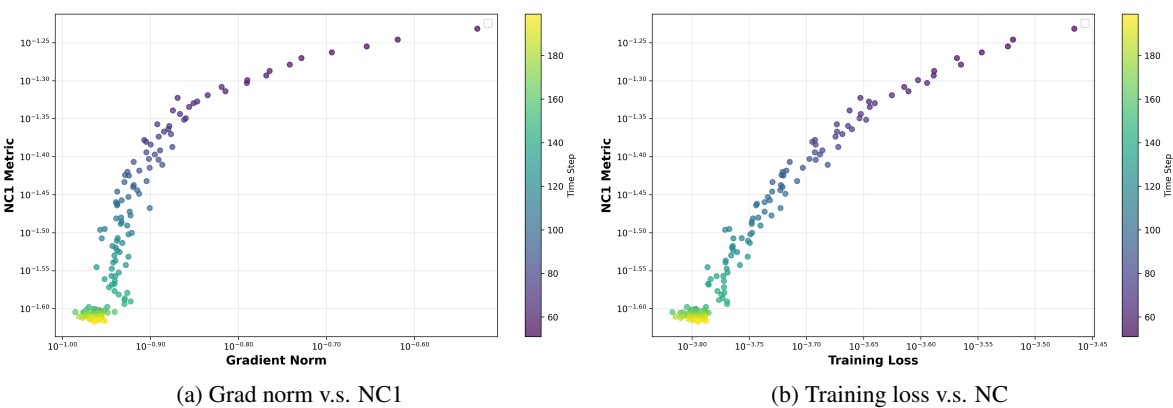

(a) Grad norm v.s. NC1

(b) Training loss v.s. NC

Figure 6: Scatterplot of average gradient norm/training loss and average NC1 for training VGG-11 on CIFAR-10. The plot starts from epoch 51, right after the learning rate is reduced by a factor 10. The color bar indicates the training time step.

