# OpenReview forum: "Neural Collapse Beyond the Unconstrained Features Model:  Landscape, Dynamics, and Generalization in the Mean-Field Regime"
_ICML.cc/2025/Conference — ICML 2025 spotlightposter_

### Official Review · Reviewer_sgNN · 2025-02-26

**Overall Recommendation:** 3

**Summary:**

This paper provides a theoretically rigorous proof on the neural collapse (NC) phenomenon with a three-layer neural network in the mean-field regime under mean-square-error (MSE) loss. It shows that under gradient flow (GF), the within-class variability (namely NC1) vanishes as the time $t$ tends to infinity. Also, it establishes an upper bound on the vanishing test error, ensuring both NC and good generalization performance can be achieved simultaneously.

**Claims And Evidence:**

The theoretical results are proven in the main text as well as in the appendix. Experiment results also show a collapse in NC1.

**Essential References Not Discussed:**

I think most related works have been discussed thoroughly in the paper.

**Experimental Designs Or Analyses:**

The experiment results match and validate the theoretical claim.

**Methods And Evaluation Criteria:**

Yes, the experiment is done on a 3-layer MLP, matching the setting in the theory. Standard datasets MNIST and CIFAR-10 are used.

**Other Comments Or Suggestions:**

I have no other comments.

**Other Strengths And Weaknesses:**

This paper does not require the weight balancedness assumptions used in [1].

The theorems and lemmata are presented and elaborated in a clear manner, which I really appreciate.

---
[1] Jacot, Arthur, et al. "Wide neural networks trained with weight decay provably exhibit neural collapse." arXiv preprint arXiv:2410.04887 (2024).

**Questions For Authors:**

I would like to ask:

1. Could you explain more about the difference between the result from this paper and [1]?

2. Related to [1] where they emphasized weight decay (WD) in the title, how could one interpret the effect of WD on the main result, say Corollary 4.9?

3. Are there any technical difficulties to extend the setting to deeper and more complex networks?

4. What is the hardest part when one tries to extend the argument in the mean-field setting to other losses, say CE?


---
[1] [1] Jacot, Arthur, et al. "Wide neural networks trained with weight decay provably exhibit neural collapse." arXiv preprint arXiv:2410.04887 (2024).

**Relation To Broader Scientific Literature:**

I think the contribution of this paper is limited, when compared to broader scientific literature.

It is known that NC is very significant when the network is trained under MSE loss. In particular, it is quite intuitive that the features concentrate at their class mean when the train loss is small under MSE loss. (It would be a different story when other loss, say cross entropy loss, is considered.) Hence, the qualitative statement in the main result (corollary 4.9) is not very surprising. The statement could have been more significant when it could also quantitative, like measuring the (non-)asymptotic rate of convergence etc.

Another result (Theorem 5.3) on the vanishing test error is nicer in this sense, however, it requires the data to be almost linearly separable.

More importantly, [1] already extended the MSE NC result from unconstraint feature model (UFM) to a deeper one with several linear last layers, which is of the same goal of this paper. This paper has also discussed deeply on the comparison between [1] and their work, which I really appreciate, but I feel like the main improvement of this paper over [1] is just removing the weight balancedness condition in [1], which is only a technical requirement. Also, [1] considered networks with any depth while this paper only considered a 3-layer network.

That is why I think this paper has limited contributions related to the broader scientific literature.


---
[1] Jacot, Arthur, et al. "Wide neural networks trained with weight decay provably exhibit neural collapse." arXiv preprint arXiv:2410.04887 (2024).

**Theoretical Claims:**

I only checked the proof sketch in the main text. They seem reasonable and straightforward. Due to time limit and my lack of expertise in mean-field dynamic theory, I am not able to check all proofs from line to line in the appendix.

However, if other reviewers could point out a potential mistake in the proof, I would check it closer and participate in the discussion.

---

> ### Author Rebuttal · Authors · 2025-03-31
>
> Thank you for the valuable and detailed comments, we address concerns below.
>
> **1. Supplementary materials.**
>
> We will add this in the revision.
>
> **2. NC not surprising under MSE loss.**
>
> We politely disagree with the claim that “it is quite intuitive that the features concentrate at their class mean when the train loss is small under MSE loss.” It is true that small MSE loss implies that the output of the neural network is roughly the same for all data points in the same class. However, this is only a **necessary but not sufficient condition** for NC1 to occur. The technical subtlety is that, since the feature dimension is larger than the number of classes, there might be components of the features in the space orthogonal to the last linear layer. **Without the balancedness condition**, it is not obvious to show that these orthogonal components vanish at stationary points. Intuitively, these components make the weights more isotropic, thus they could potentially decrease the entropic penalty, but increase the L2 penalty, meaning that the tradeoff needs to be carefully considered. Thus, our results in Section 4.2 require a suitable characterization of the learned feature and careful manipulations to show that the orthogonal components are indeed vanishing with training loss and gradient norm.
>
> **3. Extension to CE loss.**
>
> Please see point 1 of our response to Reviewer g3N9.
>
> **4. Measuring rate of convergence.**
>
> We provide a convergence rate for the loss in Equation (24). This is already non-trivial given the non-convex nature of the problem and to our best knowledge, no explicit rate of convergence is known in the mean-field regime beyond the 2-layer case (which has a convex free energy). Characterizing the convergence rate of the gradient norm is harder, as it requires controlling the curvature along the whole trajectory.
>
> **5. Linear separability assumption in Theorem 5.3.**
>
> In general, it is not true that NC1 and vanishing test error co-occur under MSE loss without any assumptions on the data distribution. In fact, the occurrence of NC1 implies that the model overfits the data, and overfitting is not always benign without additional assumptions: classic benign overfitting results for simple models such as linear regression require a decay in the data spectrum and proper alignment between data spectrum and signal. In this sense, showing the co-occurrence of NC1 and vanishing test error may be regarded as a harder problem than benign overfitting, and the latter phenomenon has not been investigated in the mean-field regime to our best knowledge.
>
> **6. Comparison of our work to [1].**
>
> First, [1] operates in the NTK regime where the behavior of the weights is fundamentally different from the mean-field regime that we consider. More precisely, the proof of Theorem 4.4 in [1] distinguishes two phases in training: in the first phase, the loss becomes small and no feature learning takes place; in the second phase, the weight matrices in the linear part of the network become balanced and NC1 occurs. In contrast, our work operates in the mean-field regime where feature learning takes place from the beginning and, therefore, there is no need for a second phase of the dynamics: NC1 occurs as soon as the loss and its gradient are small. Our experiments on ResNet-18 and VGG-11 also indicate this relation (see point 1 of our response to Reviewer AaqF for more details).
>
>
> Second, the balancedness condition in [1] is not just a technical requirement, but the key reason for the occurrence of NC1. In fact, the balancedness condition rules out the possibility of having any components of the features in the orthogonal space to the last linear layer. However, our results both theoretical (see Lemma 4.5) and experimental (see the end of Section 4.2) conclusively demonstrate that NC1 holds even if the network is not balanced.
>
> **7. Impact of weight decay.**
>
> Firstly, non-zero weight decay is necessary to ensure non-vanishing $W$ and features at all stationary points. Secondly, the weight decay affects the order of the empirical loss we converge to: when $\lambda_\rho = \beta^{-1}$, the empirical loss at convergence is of order $\beta^{-1}$. This is reflected in Corollary 4.9, as better NC1 guarantees (smaller $\delta_0$) require smaller weight decay $\beta^{-1} \leq poly(\delta_0).$
>
> **8. Extension to deeper/more complex networks.**
>
> We expect that our results can be extended to a deep network with many linear layers, see point 4 of our response to Reviewer AaqF. In contrast, analyzing deep networks with many nonlinear layers would require fundamentally different techniques and we regard it as an exciting future direction. Towards this goal, two possible strategies are (i) to characterize features at an $\epsilon_S$-stationary point, or (ii) to directly consider the evolution of features during training.
>
> [1] Jacot, Sukenik, Wang, Mondelli, Wide neural networks trained with weight decay provably exhibit neural collapse.

---

> > ### Comment · Reviewer_sgNN · 2025-04-05
> >
> > Thank you for the detailed answer. Now I have a better understanding of the contribution of the paper.
> >
> > I would like to keep my original scoring. And I wish the authors the best for the remaining review period.

---

### Official Review · Reviewer_AaqF · 2025-03-09

**Overall Recommendation:** 2

**Summary:**

The authors study the how a certain aspect of neural collapse (NC) - namely, within-class variability tending to zero, can be provably associated with convergence to nearly stationary points of the loss function in noisy gradient flow dynamics for a 3 layer neural network, where the last two layers have linear activations.

**Claims And Evidence:**

The claims made in the submission seem to be well supported by formal proofs.

**Essential References Not Discussed:**

None that I am aware of.

**Experimental Designs Or Analyses:**

I've checked the soundness of the experiments given, and found them to be lacking, in the sense that the experiments do not verify the claims made in the paper.
While Fig.1 shows a qualitative behavior that aligns with the statement that NC1, gradient norm, and the empirical loss all converge to small values together, they do not constitute a sufficient support for these claims in practice. Two things can be done when providing empirical results for a theory paper:
1) All statements are exact under the assumptions made, and no further verification is needed, aside from perhaps a numerical simulation that shows, for instance, that in a completely controlled setting, SGD on the 3 layer network with linearly separable data (say very clustered gaussian mixture) converges to NC1 with predicted dependence on the training/test loss that is given by gradient flow on the free energy.
2) An attempt to make a broad statement connected to real networks beyond the setting studied, not knowing all the constant factors, but showing that the relation between NC1/train loss/gradient norm is maintained by looking at quantities that scale with (for example the $\beta$ parameter) by performing a scan over many values.

This paper chose neither, and only shows a qualitative result, that I feel does not really contribute much, given the theoretical results.

**Methods And Evaluation Criteria:**

Since the paper is mainly theoretical in nature, the claims relate to bounds and convergence properties. The evaluation criteria are therefore mainly the voracity of the proofs. The empirical evidence are only meant to show that even imbalanced networks tend towards small loss/gradient norm and NC1 together, and for that they roughly make sense.

**Other Comments Or Suggestions:**

I did not find any glaring typos.

**Other Strengths And Weaknesses:**

**Strengths**:

- The paper is well written and fairly easy to follow, apart from minor clarifications that might be useful to see.
- The theoretical results seem to be correct.

The main contributions are extending previous results. In particular, under mild assumptions of small gradient norm and small loss, the authors show that a 3-layer network with the last two layers being linear converge under gradient flow to NC1.

This relaxes assumptions from previous works, specifically:

1. Remove the balancedness assumption on all the linear layers and show NC1 occurs even for unbalanced layers

2. Mean field in the regime of only the first layer width tending to infinity, while the width of the second layer remains of constant order.

They further show that for linearly separable (and bounded) data the test error decreases in correspondence with NC1 convergence.

These results indeed shed further light on the phenomenon of neural collapse, and attempts to connect it further with generalization performance.

**Weaknesses:**

1. As opposed to previous works (Jacot et al, Sukenık et al.) there is no discussion regarding NC2 and NC3.

2. It seems that the major difference between this and previous works is the extension of the two linear layer setting to the SGD case, where noise induces an effective entropy regularization term in the dynamics.

3. The connection made between the theoretical results and real-world networks is not studied seriously, but seems a bit of an afterthought.

The combination of these weaknesses makes the contribution, in my opinion, overly incremental.

**Questions For Authors:**

116 - What does Law(a,u) mean? it is never specified clearly.

112 - I think the phrasing implies you introduce an entropy regularization term but it comes simply from performing SGD on the predefined loss, so the dynamics are done on a free energy which includes the entropy penalty term, right?

Does condition A3 mean the results don’t hold for ReLU? how would they change?

**Relation To Broader Scientific Literature:**

The results of these authors relate to the broad phenomenon of neural collapse which is observed in trained classifiers  across different networks and datasets, and attempts to connect a specific NC property in a tractable setup with the loss landscape both at training and test, thus relating NC to generalization performance, which is of broad interest.

**Theoretical Claims:**

I checked the main theorems given in the paper, but did not re-derive them.

---

> ### Author Rebuttal · Authors · 2025-03-31
>
> Thank you for the detailed review. We address concerns below.
>
> **1. Lack of soundness in experimental design.**
>
> We followed the suggestions and did additional experiments, please download pdf in https://github.com/conferenceanonymous152/icml25
>
> (a) For three-layer networks, Figure 3 shows the linear relation between log of NC1 and log of gradient norm/training loss throughout training. Different points correspond to different epochs of training (purple ones in the top-right correspond to early stages, and yellow ones in the bottom-left correspond to late stages), and we average the results of 4 independent experiments. The plots indicate a polynomial relation between the quantities as predicted by Equation (17).
>
> (b) For practical models, we show that ResNet-18 and VGG-11 trained on CIFAR-10 exhibit a similar behavior as three-layer networks. Figure 4 is similar to Figure 1 in the paper. The scatterplots in Figures 5, 6 plot the $\log-\log$ relation between NC1 and gradient norm/training loss, and they indicate a polynomial relation in the terminal phase of training (while the early phase of training may be a little different).
>
> **2. NC2-NC3.**
>
> To prove NC2-NC3, [1] uses that the number of linear layers is large. If we stack many linear layers on top of the feature-learning layer and run gradient flow, we expect that NC2-NC3, as well as NC1, can be shown. As for NC1, our approach could be adapted as follows. Let the network be $f(x; \rho, W_2, \dots, W_L) = W_L \dots W_2 h_\rho(x),$ and define $ W^\top =W_L \dots W_2$. Then, the characterization in Theorem 4.2 still holds, since the condition of Definition 4.1 only involves the gradient computed over $\rho$. The lower bound in Lemma 4.3 holds and the upper bound becomes $\sigma_{max}(W) \leq O( (\log \beta)^L ) = o(\beta)$, which implies Corollary 4.4. Theorem 4.8 can be proved by letting  $t_*$ be the minimal first hitting time for $W_2, \dots, W_L$ and $\rho$ and following a similar strategy. Finally, since approximate NC1 holds for $H_\rho(x),$ it also holds for $W^\top H_\rho(x)$, for any $W$ satisfying Lemma 4.3.
>
> As for NC2-NC3, we note that $W_2, \dots, W_L$ are balanced for all stationary points $(\rho, W_2, \dots, W_L)$. Thus, if the condition number of $W$ is bounded by a constant independent of $L$ (as in Theorem 3.1 of [1]), one can show NC2-NC3 following the approach in [1]. We note that **$W_1 := \mathbb{E}_\rho[a a^\top]$ is not balanced with $W_2, \dots, W_L$** due to Lemma 4.5 of our work. Thus, to show NC1 for a network with many linear layers, the results of [1] cannot be applied.
>
> **3. Major differences compared to previous work, additional entropic regularization term and question about line 112.**
>
> The entropic regularization term is not implicitly induced by SGD, but it is explicitly added in the form of a Brownian noise. In fact, we consider noisy gradient flow for training, i.e., mean-field Langevin dynamics. This is standard in the mean-field regime [2, 3], and noisy gradient flow is the limiting dynamics of noisy SGD with step size $\eta$ and any batch size (including full-batch) on networks of width-$N$ as $\eta \to 0$ and $N \to \infty$ [3].
>
> What distinguishes our work is that we show NC1 for a network that (i) is trained end-to-end and (ii) exhibits feature learning throughout training. In contrast, previous work focuses either on UFM (which does not capture the effect of data) or on a dynamics reaching near-interpolation in the NTK regime [1] (which does not exhibit feature learning [4]). Furthermore, the occurrence of NC1 in [1] relies on the balancedness of all linear layers, which may not hold, see Lemma 4.5. Instead, our result only needs small training loss and gradient norm, which is commonly satisfied as long as the training dynamics converges. Finally, we show for the first time the co-occurrence of NC1 and vanishing test error for a class of data distributions. For additional comparison of our work to [1], please see point 6 of the response to Reviewer sgNN.
>
> **4. Connection between theoretical results and real-world networks.**
>
> See point 1.
>
> **5. Line 116.**
>
> $Law(a,u)$ means the joint measure of random variables $a,u.$
>
> **6. Assumptions (A3).**
>
> ReLU does not satisfy (A3), but we still expect our results to hold by taking the limit of a sequence of approximations to ReLU. We note that (A3) is purely technical (it ensures the well-posedness of the Wasserstein gradient flow) and standard in works [2, 3] considering the mean-field regime.
>
> [1] Jacot, Sukenik, Wang, Mondelli, Wide neural networks trained with weight decay provably exhibit neural collapse.
>
> [2] Suzuki, Wu, Nitanda, Mean-field langevin dynamics: Time-space discretization, stochastic gradient, and variance reduction.
>
> [3] Mei, Montanari, Nguyen, A mean field view of the landscape of two-layer neural networks.
>
> [4] Yang, Hu, Tensor Programs IV: Feature Learning in Infinite-Width Neural Networks.

---

### Official Review · Reviewer_g3N9 · 2025-03-10

**Overall Recommendation:** 4

**Summary:**

The paper theoretically studies the phenomenon of neural collapse in classification, focusing on its most basic property: the vanishing of within-class variability. Unlike data-agnostic prior work analysing the unconstrained features model (UFO), this work adopts a data-specific perspective by considering a three-layer neural network with a feature-learning component in the mean-field regime. The authors establish a connection between NC1 and the loss landscape, proving that points with small empirical loss and gradient norm are roughly NC1 solutions. They also demonstrate that gradient flow dynamics on the three-layer model converges to NC1 solutions. Finally, they show that NC1 and vanishing test error can coexist for well-separated data distribution, connecting neural collapse to generalisation.

**Update after rebuttal**

I thank the authors for the detailed response. I consider the paper a solid and relevant contribution, so I recommend its acceptance.

**Claims And Evidence:**

Although I have not thoroughly checked the proofs, the claims seem reasonable and supported by rigorous mathematical analysis. The authors make use of common assumptions in the literature, present proof sketches and a fair discussion of related work.

**Essential References Not Discussed:**

I am unaware of any essential references that were not discussed in the paper.

**Experimental Designs Or Analyses:**

The paper presents comprehensive numerical experiments using MNIST and CIFAR-100 to validate the theoretical results. The experiments are well-designed, and the results are consistent with the theoretical claims.

**Methods And Evaluation Criteria:**

The methods and evaluation criteria make sense for the theoretical analysis conducted in this work.

**Other Comments Or Suggestions:**

I believe there is a typo in the paper's title: "Unconstrainted" should be "Unconstrained".

**Other Strengths And Weaknesses:**

Strengths:

- The paper bridges neural collapse to loss landscape, gradient flow dynamics and generalisation. It presents a rigorous and fairly general analysis.

- The idea of taking a two-layer neural network in the mean field regime and stacking a linear layer is quite interesting.

- The paper is well-written, and the results are well-stated. It is also well-motivated (going beyond UFO) and clearly tries to review, connect and extend previous literature.

Weaknesses:

- It is probably not feasible analytically, but a small remark on the difficulties of extending the results to cross-entropy loss would be interesting. I believe the mean-field regime has been explored only for the squared loss, which would be a complication for such an analysis (see questions for authors).

**Questions For Authors:**

1)  If a similar analysis for cross-entropy loss were possible, would the authors expect drastic changes in the qualitative picture?

2) Could you clarify the technical reason for considering the approximated model truncating the second layer in Section 5? Would the following conclusions be affected if the full model was considered?

**Relation To Broader Scientific Literature:**

The paper seems to be well-positioned within the broader scientific literature. The authors provided a comprehensive review of the related work and are very clear about the difference between their work and previous work. They also highlight the fact that their work can offer new insights into gradient-based optimization in deep learning.

**Theoretical Claims:**

I did not check the proofs in detail.

---

> ### Author Rebuttal · Authors · 2025-03-31
>
> Thank you for your valuable review and positive feedback. We address questions and concerns below.
>
> **1. Difficulty in extending results to CE loss, and expected qualitative results.**
>
> Thank you for the question. It is indeed interesting to extend our results to CE loss and we summarize two major difficulties below:
>
> * The intuitive difference between MSE loss and CE loss is that small MSE loss implies that the output of the neural network is roughly the same for all data points in the same class, as the label of data points in the same class is the same. In our model, we stack a linear layer after the feature learning part, which implies that data points in the same class having approximately the same output is a **necessary but not sufficient condition** for NC1 to happen. To see this, suppose $h(x_1), h(x_2)$ are the features of two data points in the same class. Then, $||h(x_1)-h(x_2)||\_2^2 \geq  \frac{1}{\lambda_\max(W^\top W)}||W^\top h(x_1)-W^\top h(x_2)||_2^2.$ Now, in order to extend our results to CE loss, one would have to rule out the possibility that different data points in the same class could have different logits, and it is unclear how to do so in our setting even when the loss is small. However, we believe that this is just a technical issue coming from the fact that we consider a linear classifier in the last layer.
>
> * Another technical issue is that our results require the characterization of features in Equation (6). This only holds for MSE loss, and a different characterization would be needed for CE loss.
>
> That being said, we do expect a similar characterization to hold for CE loss. As a first step towards handling the CE loss, we note that there are indeed papers in the mean-field regime training with CE loss [1,2] or training with general losses that include CE [3]. However, all those papers consider two-layer networks, [1] requires no regularization terms, and [2] focuses on the specific problem of learning parities.
>
> **2. The technical reason for considering the approximated model truncating the second layer in Section 5, and whether the following conclusions would be affected if the full model was considered.**
>
> Having an approximated second layer implies that there exists a unique global optimum for the Gibbs minimizer as in Equation (27) and such a global optimum is achievable by mean-field langevin dynamics [3]. Specifically, one needs that $\nabla_\theta \frac{\delta}{\delta \rho} \mathcal{L}_n(\rho)(\theta) $ is Lipschitz in $\theta, \rho$ (cf. Proposition 2.5 in [4]), and without truncation this cannot be guaranteed. However, we note that our results in Section 5 hold uniformly for large enough $R$, which means that we also expect the same conclusion for the original model which corresponds to $R = +\infty.$ Rigorously speaking, one cannot interchange the order of the limits (in the time $t$ and the truncation parameter $R$), but we believe this to be a purely technical issue.
>
> **3. Typo in the title.**
>
> Thanks for spotting this, we will correct it in the revision.
>
> [1] Chizat, Bach, Implicit bias of gradient descent for wide two-layer neural networks trained with the logistic loss.
>
> [2] Suzuki, Wu, Oko, Nitanda, Feature learning via mean-field langevin dynamics: classifying sparse parities and beyond.
>
> [3] Suzuki, Wu, Nitanda, A. Mean-field langevin dynamics: Time-space discretization, stochastic gradient,
> and variance reduction.
>
> [4] Hu, Ren, Siska, Szpruch, Mean-field langevin dynamics and energy landscape of neural networks.

---

> > ### Comment · Reviewer_g3N9 · 2025-04-03
> >
> > I thank the authors for the detailed response. I consider the paper a solid and relevant contribution, so I recommend its acceptance.

---

### Official Review · Reviewer_GqGb · 2025-03-22

**Overall Recommendation:** 5

**Summary:**

This paper studies the emergence of variability collapse in the penultimate layer representations of a three layer mean-field NN. The authors show that points for which the gradient norm is small (approximate stationary points) also show variability collapse, and that the level of variability collapse is controlled by the gradient norm. Next, they show that gradient flow on the parameters of a three layer mean-field NN wrt MSE loss converges to approximate stationary points. Finally they consider the case of linearly separable data and show that variability collapse is also related to small test error in this case.

**Claims And Evidence:**

All of the above claims come with theoretical proofs and are convincing. The authors also consider differences between their entropy regularized free energy landscape and the landscape of L2 regularized networks, leading to solutions with imbalanced layers.

**Essential References Not Discussed:**

The discussion is comprehensive

**Experimental Designs Or Analyses:**

Yes, the experiments measuring balancedness of the solution seem to be sound.

**Methods And Evaluation Criteria:**

Yes

**Other Comments Or Suggestions:**

None

**Other Strengths And Weaknesses:**

None

**Questions For Authors:**

None

**Relation To Broader Scientific Literature:**

This paper advances the study of neural collapse in two ways - first by moving beyond the unconstrained features model they allow for considering the role of the data/kernel. Next, they consider approximate stationary points and show that variability collapse still holds. This is in contrast with prior works that typically consider stationary points where the gradient is exactly zero.

**Theoretical Claims:**

I followed the proof sketches in the main paper but did not work through the appendix.

---

> ### Author Rebuttal · Authors · 2025-03-31
>
> Thank you very much for finding that our paper advances the study of neural collapse and recommending a strong acceptance! We are happy to have a further discussion in case additional questions or comments come up.

---

### Decision · Program_Chairs · 2025-05-01

**Decision:**

Accept (spotlight poster)

**Comment:**

This paper studies the emergence of neural collapse in penultimate layer representations. Using a three-layer neural network in the mean-field regime, the authors demonstrate that (1) approximate stationary points exhibit neural collapse (i.e., within-class variability collapse), (2) gradient flow converges to these approximate stationary points, and (3) neural collapse is associated with small test error in this setting.

Overall, the paper offers a good theoretical analysis that advances our understanding of neural collapse. While some concerns have been raised regarding the limited scope—specifically, the focus on a three-layer MLP architecture in the mean-field setting with MSE loss—these restrictions may reflect the broader challenge our community faces in developing rigorous theoretical descriptions of deep learning models.